# Delay-Tolerant Local SGD for Efficient Distributed Training

## Abstract

The heavy communication for model synchronization is a major bottleneck for scaling up the distributed deep neural network training to many workers. Moreover, model synchronization can suffer from long delays in scenarios such as federated learning and geo-distributed training. Thus, it is crucial that the distributed training methods are both *delay-tolerant* AND *communication-efficient*. However, existing works cannot simultaneously address the communication delay and bandwidth constraint. To address this important and challenging problem, we propose a novel training framework $OLCO_3$ to achieve delay tolerance with a low communication budget by using stale information. $OLCO_3$ introduces novel staleness compensation and compression compensation to combat the influence of staleness and compression error. Theoretical analysis shows that $OLCO_3$ achieves the same sub-linear convergence rate as the vanilla synchronous stochastic gradient descent (SGD) method. Extensive experiments on deep learning tasks verify the effectiveness of $OLCO_3$ and its advantages over existing works.

## 1 Introduction

Data-parallel synchronous SGD is currently the workhorse algorithm for large-scale distributed deep learning tasks with many workers (*e.g.* GPUs), where each worker calculates the stochastic gradient on local data and synchronizes with the other workers in one training iteration (Goyal et al., 2017; You et al., 2017; Huo et al., 2020). However, high communication overheads make it inefficient to train large deep neural networks (DNNs) with a large number of workers. Generally speaking, the communication overheads come in two forms: 1) *high communication delay* due to the unstable network or a large number of communication hops, and 2) *large communication budget* caused by the large size of the DNN models with limited network bandwidth. Although communication delay is not a prominent problem for the data center environment, it can severely degrade training efficiency in practical scenarios, *e.g.* when the workers are geo-distributed or placed under different networks (Ethernet, cellular networks, Wi-Fi, *etc.*) in federated learning (Konečný et al., 2016).

Existing works to address the communication inefficiency of synchronous SGD can be roughly classified into three categories: 1) *pipelining* (Pipe-SGD (Li et al., 2018)); 2) *gradient compression* (Aji & Heafield, 2017; Stich et al., 2018; Alistarh et al., 2018; Yu et al., 2018; Vogels et al., 2019); and 3) *periodic averaging* (also known as Local SGD) (Stich, 2019; Lin et al., 2018a). In pipelining, the model update uses stale information such that the next iteration does not wait for the synchronization of information in the current iteration to update the model. As the synchronization barrier is removed, pipelining can overlap computation with communication to achieve *delay tolerance*. Gradient compression reduces the amount of data transferred in each iteration by condensing the gradient with a compressor $\mathcal{C}(\cdot)$. Representative methods include scalar quantization (Alistarh et al., 2017; Wen et al., 2017; Bernstein et al., 2018), gradient sparsification (Aji & Heafield, 2017; Stich et al., 2018; Alistarh et al., 2018), and vector quantization (Yu et al., 2018; Vogels et al., 2019). Periodic averaging reduces the frequency of communication by synchronizing the workers every $p$ (larger than 1) iterations. Periodic averaging is also shown to be effective for federated learning (McMahan et al., 2017). In summary, exiting works handle the high communication delay with pipelining and use gradient compression and periodic averaging to reduce the communication budget. However, all existing methods **fail to address both**. It is also unclear how the three communication-efficient techniques introduced above can be used jointly without hurting the convergence of SGD.

Table 1: Comparison of communication-efficient methods for distributed DNN training. The period $p \in \mathbb{N}_+$ is the communication interval for periodic averaging. The staleness $s \in \mathbb{N}$ is the number of communication rounds that the information used in the model update has been outdated for. For all methods in this table, delay tolerance $\mathcal{T} = sp$.

| Methods | Overlap Comput. with Comm. | Comm. Compression | $p$ Iterations per Comm. Round | Staleness $s$ Supported |
|---|---|---|---|---|
| Gradient Compression | $\times$ | $\checkmark$ | $= 1$ | $= 0$ |
| Periodic Averaging (Local SGD) | $\times$ | $\times$ | $\geq 1$ | $= 0$ |
| Pipelining (Pipe-SGD) | $\checkmark$ | $\times$ | $= 1$ | $\geq 1$ |
| CoCoD-SGD | $\checkmark$ | $\times$ | $\geq 1$ | $= 1$ |
| OverlapLocalSGD | $\checkmark$ | $\times$ | $\geq 1$ | $= 1$ |
| OLCO$_3$ (Ours) | $\checkmark$ | $\checkmark$ | $\geq 1$ | $\geq 1$ |

In this paper, we propose a novel framework Overlap Local Computation with Compressed Communication (*i.e.*, OLCO$_3$) to make distributed training both *delay-tolerant* AND *communication efficient* by enabling and improving the combination of the above three communication-efficient techniques. In Table 1, we compare OLCO$_3$ with the aforementioned works and two succeeding state-of-the-art delay-tolerant methods CoCoD-SGD (Shen et al., 2019) and OverlapLocalSGD (Wang et al., 2020). Under the periodic averaging framework, we use $p$ to denote the number of local SGD iterations per communication round, and $s$ to denote *the number of communication rounds* that the information used in the model update has been outdated for. Let the computation time of one SGD iteration be $T_{comput}$, then we can pipeline the communication and the computation when the communication delay time is less than $sp \cdot T_{comput}$. For simplicity, we define the delay tolerance of a method as $\mathcal{T} = sp$. Local SGD has to use up-to-date information for the model update ($s = 0, p \geq 1, \mathcal{T} = sp = 0$). CoCoD-SGD and OverlapLocalSGD combine pipelining and periodic averaging by using stale results from last communication round ($s = 1, p \geq 1, \mathcal{T} = sp = p$), while our OLCO$_3$ supports various staleness ($s \geq 1, p \geq 1, \mathcal{T} = sp$) and all other features in Table 1. The main contributions of this paper are summarized as follows:

- We propose the novel OLCO$_3$ method, which achieves extreme communication efficiency by addressing both the high communication delay and large communication budget issues.
- OLCO$_3$ introduces novel staleness compensation and compression compensation techniques. Convergence analysis shows that OLCO$_3$ achieves the same convergence rate as SGD.
- Extensive experiments on deep learning tasks show that OLCO$_3$ significantly outperforms existing delay-tolerant methods in both the communication efficiency and model accuracy.

## 2 BACKGROUNDS & RELATED WORKS

**SGD and Pipelining.** In distributed training, we minimize the global loss function $f(\cdot) = \frac{1}{K} \sum_{k=1}^{K} f_k(\cdot)$, where $f_k(\cdot)$ is the local loss function at worker $k \in [K]$. At iteration $t$, vanilla synchronous SGD updates the model $\mathbf{x}_t \in \mathbb{R}^d$ with learning rate $\eta_t$ via $\mathbf{x}_{t+1} = \mathbf{x}_t - \frac{\eta_t}{K} \sum_{k=1}^{K} \nabla F_k(\mathbf{x}_t; \xi_t^{(k)})$, where $\xi_t^{(k)}$ is the stochastic sampling variable and $\nabla F_k(\mathbf{x}_t; \xi_t^{(k)})$ is the corresponding stochastic gradient at worker $k$. Throughout this paper, we assume that the stochastic gradient is an unbiased estimator by default, *i.e.*, $\mathbb{E}_{\xi_t^{(k)}} \nabla F_k(\mathbf{x}_t; \xi_t^{(k)}) = \nabla f_k(\mathbf{x}_t)$.

Pipe-SGD (Li et al., 2018) parallelizes the communication and computation of SGD via pipelining. At iteration $t$, worker $k$ computes stochastic gradient $\nabla F_k(\mathbf{x}_t; \xi_t^{(k)})$ at *current model* $\mathbf{x}_t$ and communicates to get the averaged stochastic gradient $\frac{1}{K} \sum_{k=1}^{K} \nabla F_k(\mathbf{x}_t; \xi_t^{(k)})$. Instead of waiting the communication to finish, Pipe-SGD concurrently updates the current model with stale averaged stochastic gradient via $\mathbf{x}_{t+1} = \mathbf{x}_t - \frac{\eta_t}{K} \sum_{k=1}^{K} \nabla F_k(\mathbf{x}_{t-s}; \xi_{t-s}^{(k)})$. Note that Pipe-SGD is different from asynchronous SGD (Ho et al., 2013; Lian et al., 2015) which computes stochastic gradient using *stale model* and does not parallelize the computation and communication of a worker. A problem of Pipe-SGD is that its performance deteriorates severely under high communication delay (large $s$).

**Pipelining with Periodic Averaging.** CoCoD-SGD (Shen et al., 2019) utilizes periodic averaging to reduce the number of communication rounds and parallelizes the local model update and global model averaging by concurrently conducting

$$\mathbf{x}_t = \frac{1}{K}\sum_{k=1}^{K}\mathbf{x}_t^{(k)} \quad \text{and} \quad \mathbf{x}_{t+p}^{(k)} = \mathbf{x}_t^{(k)} - \sum_{\tau=t}^{t+p-1}\eta_\tau\nabla F_k(\mathbf{x}_\tau^{(k)};\xi_\tau^{(k)})\,. \tag{1}$$

in which $\mathbf{x}_t^{(k)}$ denotes the local model at worker $k$ as the local models on different workers are no longer consistent in non-communicating iterations. When the operations in Eq. (1) finishes, the local model is updated via $\mathbf{x}_{t+p}^{(k)} \leftarrow \mathbf{x}_t + \mathbf{x}_{t+p}^{(k)} - \mathbf{x}_t^{(k)}$ and $t \leftarrow t + p$. CoCoD-SGD can tolerate delay up to $p$ SGD iterations (*i.e.*, one communication round in periodic averaging). OverlapLocalSGD (Wang et al., 2020) improves CoCoD-SGD by heuristically pulling $\mathbf{x}_{t+p}^{(k)}$ back to the $\mathbf{x}_t$ after the operations in Eq. (1) via $\mathbf{x}_{t+p}^{(k)} \leftarrow (1-\alpha)\mathbf{x}_{t+p}^{(k)} + \alpha\mathbf{x}_t$ where $0 \leq \alpha < 1$. The motivation is to reduce the inconsistency in the local models across workers. OverlapLocalSGD also develops a momentum variant, which maintains a slow momentum buffer for $\mathbf{x}_t$ following SlowMo (Wang et al., 2019). As both CoCoD-SGD and OverlapLocalSGD communicates the non-compressed local model update, they suffer from a large communication budget in each communication round.

**Gradient Compression.** The gradient vector $\mathbf{v} \in \mathbb{R}^d$ can be sent with a much smaller communication budget by applying a compressor $\mathcal{C}(\cdot)$. Specifically, Scalar quantization rounds 32-bit floating-point gradient components to low-precision values of only several bits. One important such algorithm is scaled SignSGD (called SignSGD in this paper) (Bernstein et al., 2018; Karimireddy et al., 2019) which uses $\mathcal{C}(\mathbf{v}) = \frac{\|\mathbf{v}\|_1}{d}\operatorname{sign}(\mathbf{v})$ to compress $\mathbf{v}$ to 1 bit. Gradient sparsification only communicates large gradient components. Vector quantization uses a codebook where each code is a vector and quantizes the gradient vector as a linear combination of the vector codes. With the local error feedback technique (Seide et al., 2014; Lin et al., 2018b; Wu et al., 2018; Karimireddy et al., 2019; Zheng et al., 2019), which adds the previous compression error (*i.e.*, $\mathbf{v} - \mathcal{C}(\mathbf{v})$) to current gradient before compression, gradient compression can achieve comparable performance as full-precision training. Local error feedback also works for both *one-way compression* (compress the communication from worker to sever) (Karimireddy et al., 2019) and *two-way compression* (compress the communication between worker and server) (Zheng et al., 2019).

**Challenges.** Simultaneously achieving communication compression with pipelining and periodic averaging requires careful algorithm design because 1) pipelining introduces staleness, and 2) state-of-the-art vector quantization methods usually require an additional round of communication to solve the compressor $\mathcal{C}(\cdot)$, which is unfavorable in high communication delay scenarios.

## 3 THE PROPOSED FRAMEWORK: OLCO₃

In this section, we will introduce our new delay-tolerant and communication-efficient training framework OLCO₃. We discuss two variants of OLCO₃: *OLCO₃-TC* for two-way compression in master-slave communication mode, and *OLCO₃-VQ* adopting commutative vector quantization for both the master-slave and ring all-reduce communication modes. Note that one-way compression in just a special case of OLCO₃-TC and we omit it for conciseness. We use "line x" to refer to the x-th line of Algorithm 1. The key differences between OLCO₃-TC and OLCO₃-VQ are marked in red color.

### 3.1 OLCO₃-TC FOR TWO-WAY COMPRESSION

**Motivation.** OLCO₃-TC is presented in the green part of Algorithm 1 for efficient master-slave distributed training. Naively pipelining local computation with compressed communication will break the update rule of momentum SGD for the averaged model $\mathbf{x}_t = \frac{1}{K}\sum_{k=1}^{K}\mathbf{x}_t^{(k)}$, leading to non-convergence. Therefore, we consider an *auxiliary variable* $\tilde{\mathbf{x}}_t := \frac{1}{K}\sum_{k=1}^{K}\mathbf{x}_t^{(k)} - \frac{1}{K}\sum_{k=1}^{K}\mathbf{e}_t^{(k)} - \mathbf{e}_t$, where $\mathbf{e}_t^{(k)}$ is the local compression error at worker $k$ and $\mathbf{e}_t$ is the compression error at the server. If $\tilde{\mathbf{x}}_t$ can follow the update rule of momentum SGD, then the real trained model $\mathbf{x}_t$ will gradually approach $\tilde{\mathbf{x}}_t$ as the training converges because the gradient and errors $\mathbf{e}_t^{(k)}, \mathbf{e}_t \to \mathbf{0}$.

**Pipelining.** For non-communicating iterations, we perform the local update following Local SGD (line 4). A communicating iteration takes place every $p$ iterations. To pipeline the communication

**Algorithm 1** Overlap Local Computation with Compressed Communication (OLCO$_3$) on worker $k \in [K]$. Green part: OLCO$_3$-TC; Yellow part: OLCO$_3$-VQ. Best view in color.

1: **Input:** period $p \geq 1$, staleness $s \geq 0$, number of iterations $T$, number of workers $K$, learning rate $\{\eta_t\}_{t=0}^{T-1}$, and compression scheme $\mathcal{C}(\cdot)$.

2: **Initialize:** Local model $\mathbf{x}_0^{(k)} = \mathbf{x}_0$, local error $\mathbf{e}_0^{(k)} = \mathbf{0}$, server error $\mathbf{e}_0 = \mathbf{0}$, local momentum buffer $\mathbf{m}_0^{(k)} = \mathbf{0}$, and momentum constant $0 < \mu < 1$. Variables with negative subscripts are $\mathbf{0}$.

3: **for** $t = 0, 1, \cdots, T - 1$ **do**

4:      $\mathbf{m}_{t+1}^{(k)} = \mu\mathbf{m}_t^{(k)} + \nabla F_k(\mathbf{x}_t^{(k)}; \xi_t^{(k)})$, $\mathbf{x}_{t+1}^{(k)} = \mathbf{x}_t^{(k)} - \eta_t\mathbf{m}_{t+1}^{(k)}$      *// Momentum Local SGD.*

5:      **if** $(t + 1) \mod p = 0$ **then**

6:          Maintain or reset the momentum buffer.

7:          $\Delta_{t+1}^{(k)} = \mathbf{x}_{t+1-p}^{(k)} - \mathbf{x}_{t+1}^{(k)} + \mathbf{e}_t^{(k)}$      *// Compression compensation.*

8:          $\mathbf{e}_{t+1}^{(k)} = \mathbf{e}_{t+2}^{(k)} = \cdots = \mathbf{e}_{t+p}^{(k)} = \Delta_{t+1}^{(k)} - \mathcal{C}(\Delta_{t+1}^{(k)})$      *// Compression.*

9:          Invoke the communication thread *in parallel* which does:

10:              (1) Send $\mathcal{C}(\Delta_{t+1}^{(k)})$ to and receive $\mathcal{C}(\Delta_{t+1})$ from the server node.

11:              (2) Server: $\Delta_{t+1} = \frac{1}{K}\sum_{k=1}^K \mathcal{C}(\Delta_{t+1}^{(k)}) + \mathbf{e}_t$; $\mathbf{e}_{t+1} = \mathbf{e}_{t+2} = \cdots = \mathbf{e}_{t+p} = \Delta_{t+1} - \mathcal{C}(\Delta_{t+1})$.

12:          Block until $\mathcal{C}(\Delta_{t+1-sp})$ is ready.

13:          $\mathbf{x}_{t+1} = \mathbf{x}_{t+1-p} - \mathcal{C}(\Delta_{t+1-sp})$

14:          $\mathbf{x}_{t+1}^{(k)} \leftarrow \mathbf{x}_{t+1} - \sum_{i=0}^{s-1} \mathcal{C}(\Delta_{t+1-ip}^{(k)})$      *// Staleness compensation.*

15:          $\Delta_{t+1}^{(k)} = \mathbf{x}_{t+1-p}^{(k)} - \mathbf{x}_{t+1}^{(k)} + \mathbf{e}_{t-sp}^{(k)}$      *// Compression compensation.*

16:          Invoke the communication thread *in parallel* which does:

17:              (1) $\mathbf{e}_{t+1}^{(k)} = \mathbf{e}_{t+2}^{(k)} = \cdots = \mathbf{e}_{t+p}^{(k)} = \Delta_{t+1}^{(k)} - \mathcal{C}(\Delta_{t+1}^{(k)})$      *// Compression.*

18:              (2) Average $\frac{1}{K}\sum_{k=1}^K \mathcal{C}(\Delta_{t+1}^{(k)})$ by ring all-reduce or master-slave communication.

19:          Block until $\frac{1}{K}\sum_{k=1}^K \mathcal{C}(\Delta_{t+1-sp}^{(k)})$ and $\mathbf{e}_{t+1-sp}^{(k)}$ is ready.

20:          $\mathbf{x}_{t+1} = \mathbf{x}_{t+1-p} - \frac{1}{K}\sum_{k=1}^K \mathcal{C}(\Delta_{t+1-sp}^{(k)})$

21:          $\mathbf{x}_{t+1}^{(k)} \leftarrow \mathbf{x}_{t+1} - \sum_{i=0}^{s-1} \Delta_{t+1-ip}^{(k)}$      *// Staleness compensation.*

22:      **end if**

23: **end for**

24: **Output:** averaged model $\mathbf{x}_T = \frac{1}{K}\sum_{k=1}^K \mathbf{x}_T^{(k)}$

and computation, we compress the local update $\Delta_{t+1}^{(k)}$ (line 7) for efficient communication, and at the same time, try to update the model with a *stale compressed global update* $\mathcal{C}(\Delta_{t+1-sp})$ (line 13) that has been outdated for $s$ communication rounds (*i.e.*, the staleness is $s$). The momentum buffer can be maintained or reset to zero every $p$ iteration (line 6). If the delay tolerance $\mathcal{T} = sp$ is larger than the actual communication delay, the blocking in line 12 becomes a no-op and there will be no synchronization barrier. The server compresses the sum of the compressed local updates from all workers (line 11) and sends it back, making OLCO$_3$-TC an efficient two-way compression method.

**Compensation.** To make the update of the auxiliary variable $\tilde{\mathbf{x}}_t$ follow momentum SGD, we propose to 1) compensate staleness with all *compressed local updates* with staleness $\in [0, s - 1]$ (line 14), which requires no communication and allows less stale local update to affect the local model, and 2) maintain a local error (line 8) and add it to the next local update before compression (line 7) to compensate the compression error. With the two compensation techniques in OLCO$_3$-TC, Lemma 1 shows that the update rule of $\tilde{\mathbf{x}}_t$ follows momentum SGD with averaged momentum $\frac{1}{K}\sum_{k=1}^K \mathbf{m}_t^{(k)}$.

**Lemma 1.** *For OLCO$_3$-TC, let $\tilde{\mathbf{x}}_t := \frac{1}{K}\sum_{k=1}^K \mathbf{x}_t^{(k)} - \frac{1}{K}\sum_{k=1}^K \mathbf{e}_t^{(k)} - \mathbf{e}_{t-sp}$, then we have $\tilde{\mathbf{x}}_t = \tilde{\mathbf{x}}_{t-1} - \frac{\eta_{t-1}}{K}\sum_{k=1}^K \mathbf{m}_t^{(k)}$.*

Note that there is a "gradient mismatch" problem as the local momentum $\mathbf{m}_t^{(k)}$ is computed at the local model $\mathbf{x}_t^{(k)}$ but used in the update rule of the auxiliary variable $\tilde{\mathbf{x}}_t$ (Karimireddy et al., 2019; Xu et al., 2020). However, our analysis shows that it does not affect the convergence rate. We have also considered OLCO$_3$ for one-way compression (*i.e.*, OLCO$_3$-OC) as a special case of OLCO$_3$-TC. In

OLCO$_3$-OC, the compressor at the server side is identity function and the server error $\mathbf{e}_t$ is $\mathbf{0}$. For OLCO$_3$-OC, the auxiliary variable $\tilde{\mathbf{x}}_t$ also follows momentum SGD as stated in Lemma 2.

**Lemma 2.** *For OLCO$_3$-OC, let $\tilde{\boldsymbol{x}}_t \coloneqq \frac{1}{K}\sum_{k=1}^{K}\boldsymbol{x}_t^{(k)} - \frac{1}{K}\sum_{k=1}^{K}\boldsymbol{e}_t^{(k)}$, then we have $\tilde{\boldsymbol{x}}_t = \tilde{\boldsymbol{x}}_{t-1} - \frac{\eta_{t-1}}{K}\sum_{k=1}^{K}\boldsymbol{m}_t^{(k)}$.*

We can see that the delay tolerance of both OLCO$_3$-TC and OLCO$_3$-OC are $\mathcal{T} = sp(s \geq 1, p \geq 1)$. They have a memory overhead of $\mathcal{O}(sd)$ for storing information with staleness $\in [0, s-1]$. For most compression schemes such as SignSGD, the computation complexity of $\mathcal{C}(\cdot)$ is $\mathcal{O}(d)$.

## 3.2 OLCO$_3$-VQ FOR COMMUTATIVE VECTOR QUANTIZATION

OLCO$_3$-TC and OLCO$_3$-OC work for compressed communication in the master-slave communication paradigm. In contrast, OLCO$_3$-VQ (the yellow part of Algorithm 1) works for both the master-slave and ring all-reduce communication paradigms. Ring all-reduce minimizes communication congestion by shifting from centralized aggregation in master-slave communication (Yu et al., 2018). OLCO$_3$-VQ relies on a state-of-the-art vector quantization scheme, PowerSGD (Vogels et al., 2019), which satisfies commutability for compression, *i.e.*, $\mathcal{C}(\mathbf{v}_1) + \mathcal{C}(\mathbf{v}_2) = \mathcal{C}(\mathbf{v}_1 + \mathbf{v}_2)$. However, directly using PowerSGD breaks the delay tolerance of OLCO$_3$ as its compressor $\mathcal{C}(\cdot)$ needs communication and introduces synchronization barriers. Specifically, PowerSGD invokes communication across all workers to compute a transformation matrix, which is used to project the local updates to the compressed form.

**Pipelining with Communication-Dependent Compressor.** To make OLCO$_3$-VQ delay-tolerant, we further propose a novel compression compensation technique with the *stale local error* (line 15). This is in contrast to OLCO$_3$-TC and OLCO$_3$-OC, which use *immediate* compressed results to calculate the *up-to-date local error*. As this technique removes the dependency on immediate compressed results, we can move the whole compression and averaging process to the communication thread (lines 17 and 18). For staleness compensation, OLCO$_3$-VQ uses all *uncompressed local updates* with staleness $\in [0, s-1]$ instead of compressed local updates in OLCO$_3$-TC. With the two compensation techniques, Lemma 3 shows that for OLCO$_3$-VQ, the auxiliary variable $\tilde{\mathbf{x}}_t$ associated with the *stale local error* also follows the momentum SGD update rule.

**Lemma 3.** *For OLCO$_3$-VQ, let $\tilde{\boldsymbol{x}}_t \coloneqq \frac{1}{K}\sum_{k=1}^{K}\boldsymbol{x}_t^{(k)} - \frac{1}{K}\sum_{k=1}^{K}\boldsymbol{e}_{t-sp}^{(k)}$, then we have $\tilde{\boldsymbol{x}}_t = \tilde{\boldsymbol{x}}_{t-1} - \frac{\eta_{t-1}}{K}\sum_{k=1}^{K}\boldsymbol{m}_t^{(k)}$.*

## 4 THEORETICAL RESULTS

In this section, we provide the convergence results of the OLCO$_3$ variants for both SGD and momentum SGD maintaining momentum (line 6 of Algorithm 1) with common assumptions. As OLCO$_3$-OC is a special case of OLCO$_3$-TC, we only analyze OLCO$_3$-TC and OLCO$_3$-VQ. The detailed proofs of Theorems 1, 2, 3, and 4 can be found in Appendix D, E, F, and G respectively. The detailed proofs of Lemma 1, 2, and 3 can be found in Appendix C. We use $f_*$ to denote the optimal loss.

**Assumption 1.** *(L-Lipschitz Smoothness) Both the local ($f_k(\cdot)$) and global ($f(\cdot) = \frac{1}{K}\sum_{k=1}^{K}f_k(\cdot)$) loss functions are L-smooth, i.e.,*

$$\|\nabla f(\boldsymbol{x}) - \nabla f(\boldsymbol{y})\|_2 \leq L\|\boldsymbol{x} - \boldsymbol{y}\|_2, \forall \boldsymbol{x}, \boldsymbol{y} \in \mathbb{R}^d, \tag{2}$$

$$\|\nabla f_k(\boldsymbol{x}) - \nabla f_k(\boldsymbol{y})\|_2 \leq L\|\boldsymbol{x} - \boldsymbol{y}\|_2, \forall k \in [K], \forall \boldsymbol{x}, \boldsymbol{y} \in \mathbb{R}^d. \tag{3}$$

**Assumption 2.** *(Local Bounded Variance) The local stochastic gradient $\nabla F_k(\boldsymbol{x}; \xi)$ has a bounded variance, i.e., $\mathbb{E}_{\xi \sim \mathcal{D}_k}\|\nabla F_k(\boldsymbol{x}; \xi) - \nabla f_k(\boldsymbol{x})\|_2^2 \leq \sigma^2, \forall k \in [K], \forall \boldsymbol{x} \in \mathbb{R}^d$. Note that $\mathbb{E}_{\xi \sim \mathcal{D}_k}\nabla F_k(\boldsymbol{x}; \xi) = \nabla f_k(\boldsymbol{x})$.*

**Assumption 3.** *(Bounded Variance across Workers) The $L_2$ norm of the difference of the local and global full gradient is bounded, i.e., $\|\nabla f_k(\boldsymbol{x}) - \nabla f(\boldsymbol{x})\|_2^2 \leq \kappa^2, \forall k \in [K], \forall \boldsymbol{x} \in \mathbb{R}^d$. $\kappa = 0$ leads to i.i.d. data distributions across workers.*

**Assumption 4.** *(Bounded Full Gradient) The second moment of the global full gradient is bounded, i.e., $\|\nabla f(\boldsymbol{x})\|_2^2 \leq G^2, \forall \boldsymbol{x} \in \mathbb{R}^d$.*

**Assumption 5.** *(Karimireddy et al., 2019) The compression function $\mathcal{C}(\cdot) : \mathbb{R}^d \to \mathbb{R}$ is a $\delta$-approximate compressor for $0 < \delta \leq 1$ if for all $\boldsymbol{v} \in \mathbb{R}^d$, $\|\mathcal{C}(\boldsymbol{v}) - \boldsymbol{v}\|_2^2 \leq (1 - \delta)\|\boldsymbol{v}\|_2^2$.*

### 4.1 SGD

**Theorem 1.** *For OLCO₃-VQ with vanilla SGD and under Assumptions 1, 2, 3, 4, and 5, if the learning rate $\eta \leq \min\{\frac{1}{6L(s+1)p}, \frac{1}{9L}\}$, then*

$$\frac{1}{T}\sum_{t=0}^{T-1}\mathbb{E}\|\nabla f(\frac{1}{K}\sum_{k=1}^{K}\boldsymbol{x}_t^{(k)})\|_2^2 \leq \frac{6(f(\boldsymbol{x}_0)-f_*)}{\eta T} + \frac{9\eta L\sigma^2}{K} + 12\eta^2 L^2(s+1)p\sigma^2[1+ \tag{4}$$

$$\frac{14(1-\delta)}{\delta^2}(s+1)p] + 36\eta^2 L^2(s+1)^2 p^2\kappa^2(1+\frac{5(1-\delta)}{\delta^2}) + \frac{168(1-\delta)}{\delta^2}\eta^2 L^2(s+1)^2 p^2 G^2\,.$$

If we set the learning rate $\eta = \mathcal{O}(K^{\frac{1}{2}}T^{-\frac{1}{2}})$ and the communication interval $p = \mathcal{O}(K^{-\frac{3}{4}}T^{\frac{1}{4}}(s+1)^{-1})$, the convergence rate will be $\mathcal{O}(K^{-\frac{1}{2}}T^{-\frac{1}{2}})$. The $\mathcal{O}(K^{-\frac{1}{2}}T^{-\frac{1}{2}})$ rate is the same as synchronous SGD and Local SGD, and achieves linear speedup regarding the number of workers $K$.

**Theorem 2.** *For OLCO₃-TC with vanilla SGD and under Assumptions 1, 2, 3, 4, and 5, if the learning rate $\eta \leq \min\{\frac{1}{6L(s+1)p}, \frac{1}{9L}\}$ and let $h(\delta) = \frac{1-\delta}{\delta^2}(1+\frac{4(2-\delta)}{\delta^2})$, then*

$$\frac{1}{T}\sum_{t=0}^{T-1}\mathbb{E}\|\nabla f(\frac{1}{K}\sum_{k=1}^{K}\boldsymbol{x}_t^{(k)})\|_2^2 \leq \frac{6(f(\boldsymbol{x}_0)-f_*)}{\eta T} + \frac{9\eta L\sigma^2}{K}$$

$$+ 12\eta^2 p\sigma^2(s+1+80h(\delta)p) + 12\eta^2 p^2\kappa^2(3(s+1)^2+80h(\delta)) + 960\eta^2 p^2 G^2 h(\delta)\,. \tag{5}$$

If we set the learning rate $\eta = \mathcal{O}(K^{\frac{1}{2}}T^{-\frac{1}{2}})$ and the communication interval $p = \mathcal{O}(K^{-\frac{3}{4}}T^{\frac{1}{4}}(s+1)^{-1})$, the convergence rate will be $\mathcal{O}(K^{-\frac{1}{2}}T^{-\frac{1}{2}})$. When the data distributions across workers are *i.i.d.* (*i.e.*, $\kappa = 0$), if we choose the learning rate $\eta = \mathcal{O}(K^{\frac{1}{2}}T^{-\frac{1}{2}})$ and the communication interval $p = \min\{\mathcal{O}(K^{-\frac{3}{2}}T^{\frac{1}{2}}(s+1)^{-1}), \mathcal{O}(K^{-\frac{3}{4}}T^{\frac{1}{4}})\}$ ($p = \mathcal{O}(K^{-\frac{3}{4}}T^{\frac{1}{4}})$ for a enough large $T$) instead, the convergence rate will still be $\mathcal{O}(K^{-\frac{1}{2}}T^{-\frac{1}{2}})$.

Therefore, OLCO₃-TC can tackle a larger communication interval $p$ ($\mathcal{O}(K^{-\frac{3}{4}}T^{\frac{1}{4}})$) than OLCO₃-VQ ($\mathcal{O}(K^{-\frac{3}{4}}T^{\frac{1}{4}}(s+1)^{-1})$) in the *i.i.d.* setting. But they are the same in the *non-i.i.d.* setting.

### 4.2 MOMENTUM SGD

**Theorem 3.** *For OLCO₃-VQ with Momentum SGD and under Assumptions 1, 2, 3, 4, 5, if the learning rate $\eta \leq \min\{\frac{1-\mu}{\sqrt{72}L(s+1)p}, \frac{1-\mu}{9L}\}$ and let $g(\mu, \delta, s, p) = \frac{15}{(1-\mu)^2} + \frac{60(1-\delta)(s+1)^2 p^2}{\delta^2}$, then*

$$\frac{1}{T}\sum_{t=0}^{T-1}\mathbb{E}\|\nabla f(\frac{1}{K}\sum_{k=1}^{K}\boldsymbol{x}_t^{(k)})\|_2^2 \leq \frac{6(1-\mu)(f(\boldsymbol{x}_0)-f_*)}{\eta T} + \frac{9L\eta\sigma^2}{(1-\mu)K} \tag{6}$$

$$+ \frac{4\eta^2 L^2}{(1-\mu)^2}[(4(s+1)p+g(\mu,\delta,s,p))\sigma^2 + (12(s+1)^2 p^2+g(\mu,\delta,s,p))\kappa^2 + g(\mu,\delta,s,p)G^2]\,.$$

**Theorem 4.** *For OLCO₃-TC with Momentum SGD and under Assumptions 1, 2, 3, 4, 5, if the learning rate $\eta \leq \min\{\frac{1-\mu}{\sqrt{72}L(s+1)p}, \frac{1-\mu}{9L}\}$ and $h(\delta) = \frac{1-\delta}{\delta^2}(1+\frac{4(2-\delta)}{\delta^2})$, then*

$$\frac{1}{T}\sum_{t=0}^{T-1}\mathbb{E}\|\nabla f(\frac{1}{K}\sum_{k=1}^{K}\boldsymbol{x}_t^{(k)})\|_2^2$$

$$\leq \frac{6(1-\mu)(f(\boldsymbol{x}_0)-f_*)}{\eta T} + \frac{9L\eta\sigma^2}{(1-\mu)K} + \frac{6\eta^2 L^2}{(1-\mu)^2}[\sigma^2(\frac{9}{(1-\mu)^2}+2(s+1)p+168h(\delta)p^2)$$

$$+ \kappa^2(\frac{9}{(1-\mu)^2}+6(s+1)^2 p^2+168h(\delta)p^2) + G^2(\frac{9}{(1-\mu)^2}+168h(\delta)p^2)]\,. \tag{7}$$

The same convergence rate and communication interval $p$ are achieved as in Section 4.1.

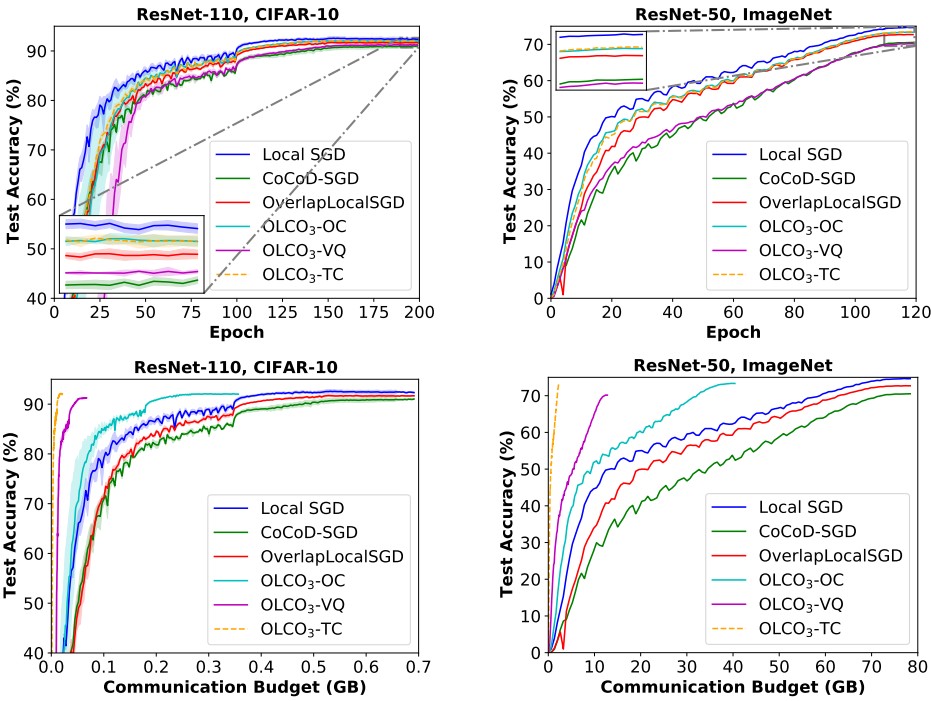

Figure 1: Training curves using $\mathcal{T} = 56$ and $s = 1$ for the delay-tolerant methods, and $\mathcal{T} = 0$ and $p = 56$ for Local SGD. Test accuracy can be found in Appendix A.3. Best viewed in color.

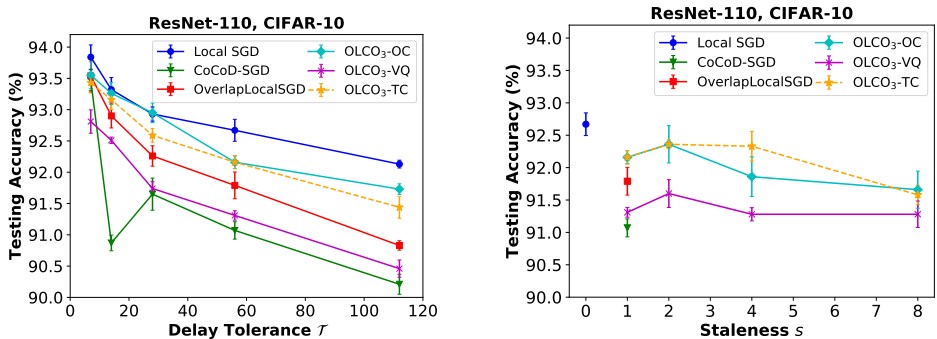

Figure 2: **Left**: Vary delay tolerance $\mathcal{T} = sp$ with staleness fixed at $s = 1$. Local SGD uses the same $p$ as the delay-tolerant methods but has $\mathcal{T} = 0$. **Right**: Vary staleness $s$ with delay tolerance fixed at $\mathcal{T} = 56$. Results for more configurations of $s$ and $p$ can be found in Appendix A.4.

## 5 EXPERIMENTS

We compare the following methods: 1) Local SGD (**baseline, NO delay tolerance** with $\mathcal{T} = 0$); 2) Pipe-SGD; 3) CoCoD-SGD; 4) OverlapLocalSGD with hyperparameters following Wang et al. (2020); 5) OLCO$_3$-OC with SignSGD compression; 6) OLCO$_3$-VQ with PowerSGD compressoin; 7) OLCO$_3$-TC with SignSGD compression. The momentum buffer is maintained (line 6 of Algorithm 1) by default. We do not report the results of Pipe-SGD as it does not converge for the large delay tolerance $\mathcal{T}$ we experimented. We train ResNet-110 (He et al., 2016) with 8 workers for the CIFAR-10 (Krizhevsky et al., 2009) image classification task, and report the mean and standard deviation of the test accuracy over 3 runs in both the *i.i.d.* and *non-i.i.d.* setting. We also train ResNet-50 with 16 workers for the ImageNet (Russakovsky et al., 2015) image classification task. More detailed descriptions of the experiment configurations can be found in Appendix A.1.

Table 2: *Non-i.i.d.* test accuracy (%) of ResNet-110 on CIFAR-10. $\mathcal{T} = 56$ for the delay-tolerant methods, and $\mathcal{T} = 0$ and $p = 56$ for Local SGD. Training curves can be found in Appendix A.2.

| Method | Delay Tolerance $\mathcal{T}$ | Compression Ratio | ResNet110 |
|---|---|---|---|
| Local SGD | 0 | 1 | $89.93 \pm 0.08$ |
| Pipe-SGD | 56 | 1 | Diverges |
| CoCoD-SGD | 56 | 1 | $87.73 \pm 0.25$ |
| OverlapLocalSGD | 56 | 1 | $88.97 \pm 0.19$ |
| OLCO$_3$-OC | 56 | 1/2 + 1/32 | $\mathbf{89.75 \pm 0.21}$ |
| OLCO$_3$-VQ | 56 | 0.0973 | $89.31 \pm 0.08$ |
| OLCO$_3$-TC | 56 | $\mathbf{1/32}$ | $\mathbf{89.73 \pm 0.25}$ |

**Delay Tolerance with Lower Communication Budget.** The training curves of ResNet-110 on CIFAR-10 and ResNet-50 on ImageNet are shown in Figure 1. We use $s = 1$ because CoCoD-SGD and OverlapLocalSGD do not support $s \geq 2$. Compared with other delay-tolerant methods, the communication budget of the OLCO$_3$ variants is significantly smaller due to compressed communication. OLCO$_3$ is also robust to communication delay with a large $\mathcal{T} = sp$. Therefore, OLCO$_3$ features extreme communication efficiency with compressed communication, delay tolerance, and low communication frequency due to periodic averaging.

**Better Model Performance.** The two plots in the first row of Figure 1 show that OLCO$_3$-OC and OLCO$_3$-TC outperforms other delay-tolerant methods and are comparable to Local SGD regarding the model accuracy. The performance of OLCO$_3$-VQ is similar to CoCoD-SGD but inferior to OverlapLocalSGD. However, in the *non-i.i.d.* results reported in Table 2, all OLCO$_3$ variants outperform existing delay-tolerant methods in accuracy. This is in line with the theoretical results in Theorems 1, 2, 3, and 4, which show that OLCO$_3$-TC can tackle a larger $p$ than OLCO$_3$-VQ in the *i.i.d.* setting but the two methods are similar in the *non-i.i.d.* setting. In the *non-i.i.d.* setting, all OLCO$_3$ variants perform very close to Local SGD. On average, OLCO$_3$-OC and OLCO$_3$-TC improve the test accuracy of CoCoD-SGD and OverlapLocalSGD by 2.0% and 0.8%, respectively. OLCO$_3$-VQ improves CoCoD-SGD and OverlapLocalSGD by 1.6% and 0.4%. These results empirically confirm that the staleness compensation and compression compensation techniques in OLCO$_3$ are effective.

**Varying Delay Tolerance.** We vary the delay tolerance $\mathcal{T}$ with staleness fixed at $s = 1$ in the left plot of Figure 2. The goal is to check the robustness of OLCO$_3$ to the different period $p$. The results show that OLCO$_3$-OC and OLCO$_3$-TC always outperform other delay-tolerant methods, and have more comparable performance to Local SGD. Note that both the OLCO$_3$-OC and OLCO$_3$-TC provide a significantly smaller communication budget according to Figure 1. OLCO$_3$-VQ also outperforms CoCoD-SGD with a much smaller communication budget.

**Varying Staleness.** We vary the staleness $s$ of OLCO$_3$ in the right plot of Figure 2 under fixed delay tolerance $\mathcal{T}$. Local SGD only supports $s = 0$ with no delay tolerance, and CoCoD-SGD and OverlapLocalSGD only support $s = 1$, so there is only one result for them in the figure. When increasing the staleness beyond 2 for OLCO$_3$, the deterioration of the model performance is very small, especially for OLCO$_3$-VQ. This suggests that the staleness compensation techniques in OLCO$_3$ are effective. The performance peaks at $s = 2$ because an appropriate staleness may introduce some noise that helps generalization. In comparison, we cannot tune staleness $s$ for better performance in CoCoD-SGD and OverlapLocalSGD.

## 6 CONCLUSION

In this work, we proposed a new OLCO$_3$ framework to achieve extreme communication efficiency with high delay tolerance and a low communication budget in distributed training. OLCO$_3$ uses novel staleness compensation and compression compensation techniques, and the theoretical results show that it converges as fast as vanilla synchronous SGD. Experimental results show that OLCO$_3$ significantly outperforms existing delay-tolerant methods in terms of the communication budget and model performance.

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

# A  ADDITIONAL EXPERIMENTAL RESULTS

## A.1  EXPERIMENTAL SETTING

All experiments are implemented with PyTorch (Paszke et al., 2019) and run on a cluster of Nvidia Tesla P40 GPUs. Each node is connected by 40Gbps Ethernet and equipped with 4 GPUs.

**CIFAR.** We train the ResNet-110 (He et al., 2016) model with 8 workers on CIFAR-10 (Krizhevsky et al., 2009) image classification task. We report the mean and standard deviation metrics over 3 runs. The base learning rate is 0.4 and the total batch size is 512. The momentum constant is 0.9 and the weight decay is $1 \times 10^{-4}$. The model is trained for 200 epochs with a learning rate decay of 0.1 at epoch 100 and 150. We linearly warm up the learning rate from 0.05 to 0.4 in the beginning 5 epochs. For $OLCO_3$ with staleness $s \in \{2, 4, 8\}$, we set the base learning rate to 0.2 due to increased staleness. The rank of PowerSGD is 4. Random cropping, random flipping, and standardization are applied as data augmentation techniques. We also train ResNet-56 to explore more combinations of $s$ and $p$ in Appendix A.4 with the same other settings.

**ImageNet.** We train the ResNet-50 model with 16 workers on ImageNet (Russakovsky et al., 2015) image classification tasks. The model is trained for 120 epochs with a cosine learning rate scheduling (Loshchilov & Hutter, 2016). The base learning rate is 0.4 and the total batch size is 2048. The momentum constant is 0.9 and the weight decay is $1 \times 10^{-4}$. We linearly warm up the learning rate from 0.025 to 0.4 in the beginning 5 epochs. The rank of PowerSGD is 50. Random cropping, random flipping, and standardization are applied as data augmentation techniques.

**The *Non-i.i.d.* Setting.** Similar to (Wang et al., 2020), we randomly choose fraction $\alpha$ of the whole data, sort the data by the class, and evenly assign them to all workers in order. For the rest fraction $(1 - \alpha)$ of the whole data, we randomly and evenly distribute them to all workers (Figure 3). When $0 < \alpha \leq 1$ is large, the data distribution across workers is *non-i.i.d* and highly skewed. When $\alpha = 0$, it becomes *i.i.d.* data distribution across workers. In our *non-i.i.d.* experiments, we choose $\alpha = 0.8$.

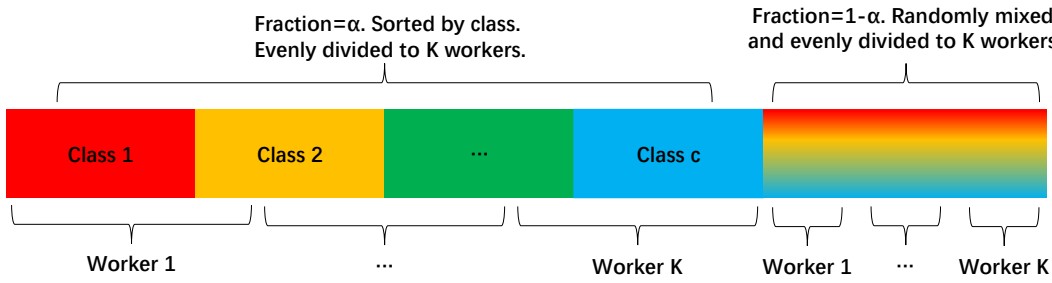

Figure 3: *Non-i.i.d.* data partition across workers. Best viewed in color.

### A.2 TRAINING CURVE

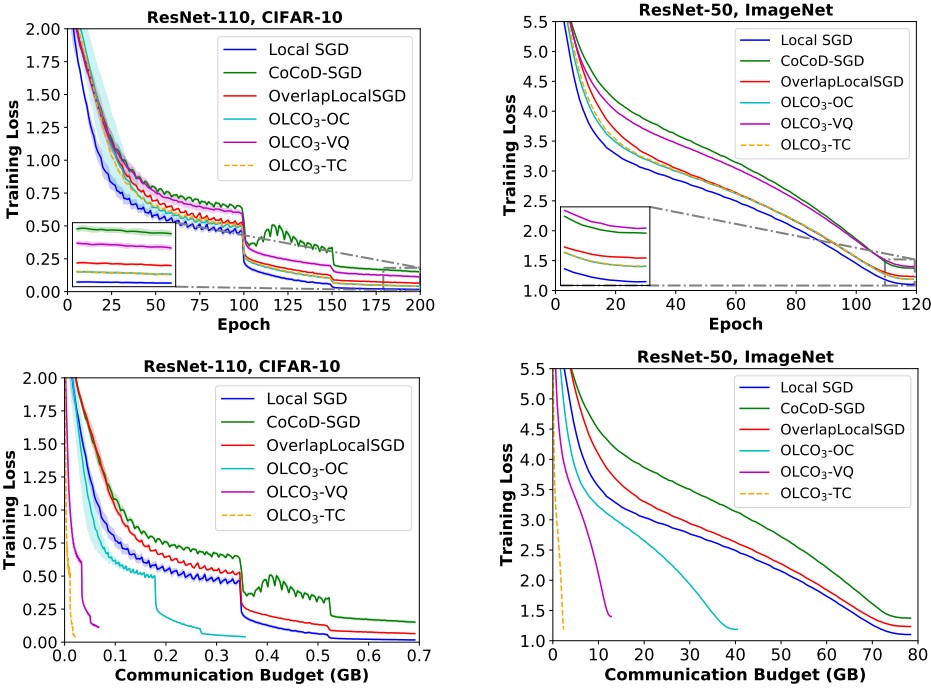

Figure 4: Training curves using $\mathcal{T} = 56$ and $s = 1$ for the delay-tolerant methods, and $\mathcal{T} = 0$ and $p = 56$ for Local SGD. Best viewed in color.

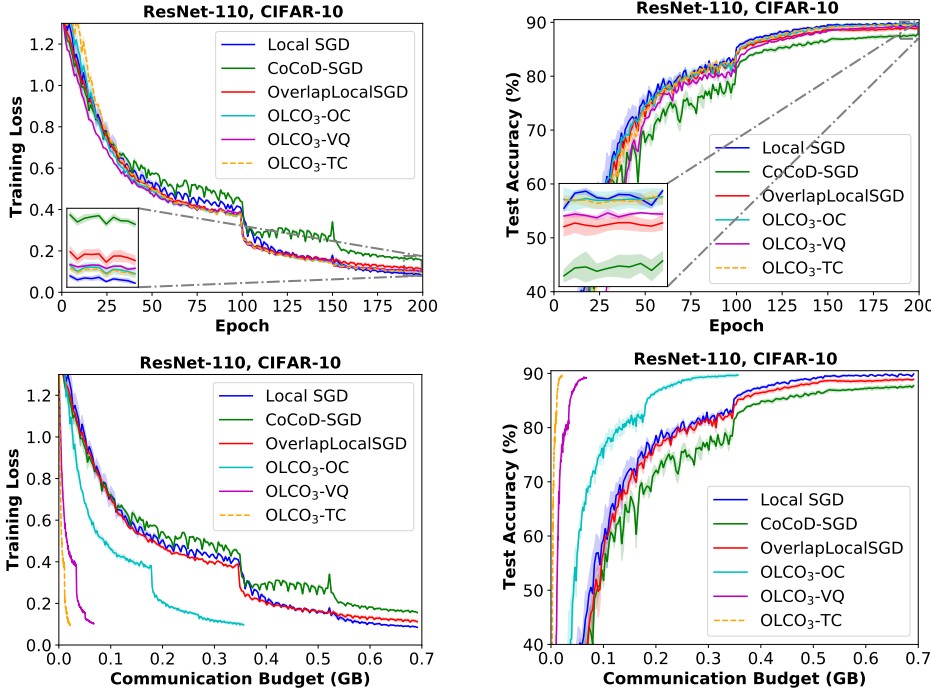

Figure 5: *Non-i.i.d.* training curves using $\mathcal{T} = 56$ and $s = 1$ for the delay-tolerant methods, and $\mathcal{T} = 0$ and $p = 56$ for Local SGD. Best viewed in color.

## A.3 TEST ACCURACY

Table 3: Test accuracy (%) of ResNet-110 on CIFAR-10 and ResNet-50 on ImageNet using $\mathcal{T} = 56$ for the delay-tolerant methods, and $\mathcal{T} = 0$ and $p = 56$ for Local SGD. CR stands for compression ratio. For each method, the first row denotes maintaining momentum and the second row denotes resetting momentum (line 6).

| Method | Delay Tolerance $\mathcal{T}$ | CR | ResNet110 | CR | ResNet-50 |
|---|---|---|---|---|---|
| Local SGD | 0 | 1 | $92.67 \pm 0.18$ 
 $92.68 \pm 0.21$ | 1 | 74.68 
 74.76 |
| CoCoD-SGD | 56 | 1 | $91.07 \pm 0.14$ 
 $90.84 \pm 0.06$ | 1 | 70.52 
 70.27 |
| OverlapLocalSGD | 56 | 1 | $91.78 \pm 0.21$ 
 $91.45 \pm 0.14$ | 1 | 72.71 
 72.87 |
| OLCO$_3$-OC | 56 | 1/2 + 1/32 | $\mathbf{92.16 \pm 0.10}$ 
 $\mathbf{92.48 \pm 0.12}$ | 1/2 + 1/32 | $\mathbf{73.36}$ 
 $\mathbf{73.40}$ |
| OLCO$_3$-VQ | 56 | 0.0973 | $91.31 \pm 0.08$ 
 $91.05 \pm 0.21$ | 0.1633 | 70.38 
 70.32 |
| OLCO$_3$-TC | 56 | $\mathbf{1/32}$ | $\mathbf{92.15 \pm 0.11}$ 
 $92.31 \pm 0.10$ | $\mathbf{1/32}$ | $\mathbf{73.49}$ 
 $73.47$ |

## A.4 HYPERPARAMETERS $s$ & $p$

Again, Figure 6 empirically confirms the theoretical results in Theorems 1, 2, 3, and 4 that OLCO$_3$-TC can handle a larger period $p$ than OLCO$_3$ and that this gap increases with the staleness $s$ in the *i.i.d.* setting. Note that in the right plot of Figure 2, the gap between OLCO$_3$-TC and OLCO$_3$-VQ does not increase with $s$ because the period $p$ is decreasing (the delay tolerance $\mathcal{T} = sp$ is fixed).

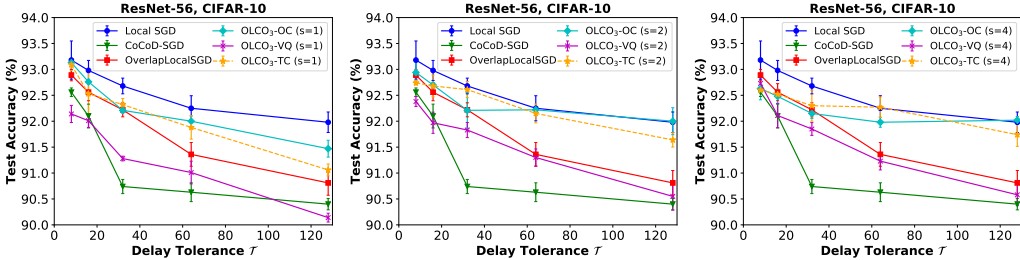

Figure 6: Vary delay tolerance $\mathcal{T}$ for ResNet-56 on CIFAR-10. We set $p$ of Local SGD equivalent to $\mathcal{T}$ of other delay-tolerant methods. **Left**: $s = 1$ for the OLCO$_3$ variants. **Middle**: $s = 2$ for the OLCO$_3$ variants. **Right**: $s = 4$ for the OLCO$_3$ variants.

Table 4: Test accuracy (%) in Figure 6 by selecting the *best configurations* of $s$ and $p$ for the OLCO$_3$ variants. We set $p$ of Local SGD equivalent to $\mathcal{T}$ of other delay-tolerant methods.

| Method | $\mathcal{T} = 8$ | $\mathcal{T} = 16$ | $\mathcal{T} = 32$ | $\mathcal{T} = 64$ | $\mathcal{T} = 128$ |
|---|---|---|---|---|---|
| Local SGD | $93.19 \pm 0.24$ | $92.98 \pm 0.19$ | $92.68 \pm 0.15$ | $92.25 \pm 0.24$ | $91.98 \pm 0.19$ |
| CoCoD-SGD | $92.56 \pm 0.08$ | $92.10 \pm 0.23$ | $90.74 \pm 0.14$ | $90.63 \pm 0.18$ | $90.40 \pm 0.11$ |
| OverlapLocalSGD | $92.89 \pm 0.11$ | $92.56 \pm 0.17$ | $92.22 \pm 0.14$ | $91.36 \pm 0.23$ | $90.81 \pm 0.24$ |
| OLCO$_3$-OC | $\mathbf{93.13 \pm 0.02}$ | $\mathbf{92.76 \pm 0.22}$ | $92.21 \pm 0.06$ | $92.00 \pm 0.10$ | $\mathbf{92.02 \pm 0.07}$ |
| OLCO$_3$-VQ | $92.73 \pm 0.12$ | $92.11 \pm 0.23$ | $91.85 \pm 0.12$ | $91.30 \pm 0.17$ | $90.58 \pm 0.21$ |
| OLCO$_3$-TC | $93.09 \pm 0.12$ | $92.68 \pm 0.21$ | $\mathbf{92.61 \pm 0.16}$ | $\mathbf{92.27 \pm 0.20}$ | $91.74 \pm 0.22$ |

## B ASSUMPTIONS

**Assumption 1.** *(L-Lipschitz Smoothness) Both the local ($f_k(\cdot)$) and global ($f(\cdot) = \frac{1}{K}\sum_{k=1}^K f_k(\cdot)$) loss functions are L-smooth, i.e.,*

$$\|\nabla f(\boldsymbol{x}) - \nabla f(\boldsymbol{y})\|_2 \le L\|\boldsymbol{x} - \boldsymbol{y}\|_2, \forall \boldsymbol{x}, \boldsymbol{y} \in \mathbb{R}^d,\tag{8}$$

$$\|\nabla f_k(\boldsymbol{x}) - \nabla f_k(\boldsymbol{y})\|_2 \le L\|\boldsymbol{x} - \boldsymbol{y}\|_2, \forall k \in [K], \forall \boldsymbol{x}, \boldsymbol{y} \in \mathbb{R}^d.\tag{9}$$

**Assumption 2.** *(Local Bounded Variance) The local stochastic gradient $\nabla F_k(\boldsymbol{x}; \xi)$ has a bounded variance, i.e.,*

$$\mathbb{E}_{\xi \sim \mathcal{D}_k} \|\nabla F_k(\boldsymbol{x}; \xi) - \nabla f_k(\boldsymbol{x})\|_2^2 \le \sigma^2, \forall k \in [K], \forall \boldsymbol{x} \in \mathbb{R}^d.\tag{10}$$

*Note that $\mathbb{E}_{\xi \sim \mathcal{D}_k} \nabla F_k(\boldsymbol{x}; \xi) = \nabla f_k(\boldsymbol{x})$.*

**Assumption 3.** *(Bounded Variance across Workers) The $L_2$ norm of the difference of the local and global full gradient is bounded, i.e.,*

$$\|\nabla f_k(\boldsymbol{x}) - \nabla f(\boldsymbol{x})\|_2^2 \le \kappa^2, \forall k \in [K], \forall \boldsymbol{x} \in \mathbb{R}^d,\tag{11}$$

*where $\kappa = 0$ leads to i.i.d. data distributions across workers.*

**Assumption 4.** *(Bounded Full Gradient) The second moment of the global full gradient is bounded, i.e.,*

$$\|\nabla f(\boldsymbol{x})\|_2^2 \le G^2, \forall \boldsymbol{x} \in \mathbb{R}^d.\tag{12}$$

**Assumption 5.** *($\delta$-approximate compressor) The compression function $\mathcal{C}(\cdot) : \mathbb{R}^d \to \mathbb{R}$ is a $\delta$-approximate compressor for $0 < \delta \le 1$ if for all $\boldsymbol{v} \in \mathbb{R}^d$,*

$$\|\mathcal{C}(\boldsymbol{v}) - \boldsymbol{v}\|_2^2 \le (1 - \delta)\|\boldsymbol{v}\|_2^2.\tag{13}$$

## C BASIC LEMMAS

**Lemma 1.** *For OLCO₃-TC, let $\tilde{\boldsymbol{x}}_t := \frac{1}{K}\sum_{k=1}^K \boldsymbol{x}_t^{(k)} - \frac{1}{K}\sum_{k=1}^K \boldsymbol{e}_t^{(k)} - \boldsymbol{e}_{t-sp}$, then we have*

$$\tilde{\boldsymbol{x}}_t = \tilde{\boldsymbol{x}}_{t-1} - \frac{\eta_{t-1}}{K}\sum_{k=1}^K \boldsymbol{m}_t^{(k)}.\tag{14}$$

*Proof.* For $t = np$ where $n$ is some integer,

$$\tilde{\mathbf{x}}_{np} = \frac{1}{K}\sum_{k=1}^K \mathbf{x}_{np}^{(k)} - \frac{1}{K}\sum_{k=1}^K \mathbf{e}_{np}^{(k)} - \mathbf{e}_{(n-s)p} = \mathbf{x}_{np} - \frac{1}{K}\sum_{k=1}^K\sum_{i=0}^{s-1} \mathcal{C}(\Delta_{(n-i)p}^{(k)}) - \frac{1}{K}\sum_{k=1}^K \mathbf{e}_{np}^{(k)} - \mathbf{e}_{(n-s)p}$$

$$= \mathbf{x}_{(n-1)p} - \mathcal{C}(\Delta_{(n-s)p}) - \frac{1}{K}\sum_{k=1}^K\sum_{i=0}^{s-1} \mathcal{C}(\Delta_{(n-i)p}^{(k)}) - \frac{1}{K}\sum_{k=1}^K \mathbf{e}_{np}^{(k)} - \mathbf{e}_{(n-s)p}$$

$$= \frac{1}{K}\sum_{k=1}^K (\mathbf{x}_{(n-1)p}^{(k)} + \sum_{i=0}^{s-1}\mathcal{C}(\Delta_{(n-1-i)p}^{(k)})) - \mathcal{C}(\Delta_{(n-s)p}) - \frac{1}{K}\sum_{k=1}^K\sum_{i=0}^{s-1}\mathcal{C}(\Delta_{(n-i)p}^{(k)}) - \frac{1}{K}\sum_{k=1}^K \mathbf{e}_{np}^{(k)} - \mathbf{e}_{(n-s)p}$$

$$= \frac{1}{K}\sum_{k=1}^K \mathbf{x}_{(n-1)p}^{(k)} - \frac{1}{K}\sum_{k=1}^K \Delta_{np}^{(k)} + \frac{1}{K}\sum_{k=1}^K \mathcal{C}(\Delta_{(n-s)p}^{(k)}) - \mathcal{C}(\Delta_{(n-s)p}) - \mathbf{e}_{(n-s)p}$$

$$= \frac{1}{K}\sum_{k=1}^K \mathbf{x}_{(n-1)p}^{(k)} - \frac{1}{K}\sum_{k=1}^K \Delta_{np}^{(k)} - \mathbf{e}_{(n-s)p-1}$$

$$= \frac{1}{K}\sum_{k=1}^K \mathbf{x}_{(n-1)p}^{(k)} - \frac{1}{K}\sum_{k=1}^K \sum_{\tau=(n-1)p}^{np-1} \eta_\tau \mathbf{m}_{\tau+1}^{(k)} - \frac{1}{K}\sum_{k=1}^K \mathbf{e}_{np-1}^{(k)} - \mathbf{e}_{(n-s)p-1} = \tilde{\mathbf{x}}_{np-1} - \frac{1}{K}\sum_{k=1}^K \eta_{np-1}\mathbf{m}_{np}^{(k)}.$$

$$\tag{15}$$

For $t \neq np$,

$$\tilde{\mathbf{x}}_t = \frac{1}{K}\sum_{k=1}^{K}\mathbf{x}_t^{(k)} - \frac{1}{K}\sum_{k=1}^{K}\mathbf{e}_t^{(k)} - \mathbf{e}_{t-sp} = \frac{1}{K}\sum_{k=1}^{K}\mathbf{x}_t^{(k)} - \frac{1}{K}\sum_{k=1}^{K}\mathbf{e}_{t-1}^{(k)} - \mathbf{e}_{t-sp-1} = \tilde{\mathbf{x}}_{t-1} - \frac{\eta_{t-1}}{K}\sum_{k=1}^{K}\mathbf{m}_t^{(k)}.$$
(16)

$\square$

**Lemma 2.** *For OLCO$_3$-OC, let $\tilde{\boldsymbol{x}}_t := \frac{1}{K}\sum_{k=1}^{K}\boldsymbol{x}_t^{(k)} - \frac{1}{K}\sum_{k=1}^{K}\boldsymbol{e}_t^{(k)}$, then we have*

$$\tilde{\boldsymbol{x}}_t = \tilde{\boldsymbol{x}}_{t-1} - \frac{\eta_{t-1}}{K}\sum_{k=1}^{K}\boldsymbol{m}_t^{(k)}.$$
(17)

*Proof.* For $t = np$ where $n$ is some integer,

$$\begin{aligned}
\tilde{\mathbf{x}}_{np} &= \frac{1}{K}\sum_{k=1}^{K}\mathbf{x}_{np}^{(k)} - \frac{1}{K}\sum_{k=1}^{K}\mathbf{e}_{np}^{(k)} = \mathbf{x}_{np} - \frac{1}{K}\sum_{k=1}^{K}\sum_{i=0}^{s-1}\mathcal{C}(\Delta_{(n-i)p}^{(k)}) - \frac{1}{K}\sum_{k=1}^{K}\mathbf{e}_{np}^{(k)} \\
&= \mathbf{x}_{(n-1)p} - \frac{1}{K}\sum_{k=1}^{K}\mathcal{C}(\Delta_{(n-s)p}^{(k)}) - \frac{1}{K}\sum_{k=1}^{K}\sum_{i=0}^{s-1}\mathcal{C}(\Delta_{(n-i)p}^{(k)}) - \frac{1}{K}\sum_{k=1}^{K}\mathbf{e}_{np}^{(k)} \\
&= \frac{1}{K}\sum_{k=1}^{K}(\mathbf{x}_{(n-1)p}^{(k)} + \sum_{i=0}^{s-1}\mathcal{C}(\Delta_{(n-1-i)p}^{(k)})) - \frac{1}{K}\sum_{k=1}^{K}\mathcal{C}(\Delta_{(n-s)p}^{(k)}) - \frac{1}{K}\sum_{k=1}^{K}\sum_{i=0}^{s-1}\mathcal{C}(\Delta_{(n-i)p}^{(k)}) - \frac{1}{K}\sum_{k=1}^{K}\mathbf{e}_{np}^{(k)} \\
&= \frac{1}{K}\sum_{k=1}^{K}\mathbf{x}_{(n-1)p}^{(k)} - \frac{1}{K}\sum_{k=1}^{K}\mathcal{C}(\Delta_{np}^{(k)}) - \frac{1}{K}\sum_{k=1}^{K}\mathbf{e}_{np}^{(k)} \\
&= \frac{1}{K}\sum_{k=1}^{K}\mathbf{x}_{(n-1)p}^{(k)} - \frac{1}{K}\sum_{k=1}^{K}\Delta_{np}^{(k)} \\
&= \frac{1}{K}\sum_{k=1}^{K}\mathbf{x}_{(n-1)p}^{(k)} - \frac{1}{K}\sum_{k=1}^{K}\sum_{\tau=(n-1)p}^{np-1}\eta_\tau\mathbf{m}_{\tau+1}^{(k)} - \frac{1}{K}\sum_{k=1}^{K}\mathbf{e}_{np-1}^{(k)} \\
&= \tilde{\mathbf{x}}_{np-1} - \frac{1}{K}\sum_{k=1}^{K}\eta_{np-1}\mathbf{m}_{np}^{(k)}.
\end{aligned}$$
(18)

For $t \neq np$,

$$\tilde{\mathbf{x}}_t = \frac{1}{K}\sum_{k=1}^{K}\mathbf{x}_t^{(k)} - \frac{1}{K}\sum_{k=1}^{K}\mathbf{e}_t^{(k)} = \frac{1}{K}\sum_{k=1}^{K}\mathbf{x}_t^{(k)} - \frac{1}{K}\sum_{k=1}^{K}\mathbf{e}_{t-1}^{(k)} = \tilde{\mathbf{x}}_{t-1} - \frac{\eta_{t-1}}{K}\sum_{k=1}^{K}\mathbf{m}_t^{(k)}.$$
(19)

$\square$

**Lemma 3.** *For OLCO$_3$-VQ, let $\tilde{\boldsymbol{x}}_t := \frac{1}{K}\sum_{k=1}^{K}\boldsymbol{x}_t^{(k)} - \frac{1}{K}\sum_{k=1}^{K}\boldsymbol{e}_{t-sp}^{(k)}$, then we have*

$$\tilde{\boldsymbol{x}}_t = \tilde{\boldsymbol{x}}_{t-1} - \frac{\eta_{t-1}}{K}\sum_{k=1}^{K}\boldsymbol{m}_t^{(k)}.$$
(20)

*Proof.* For $t = np$ where $n$ is some integer,

$$
\begin{aligned}
\tilde{\mathbf{x}}_{np} &= \frac{1}{K}\sum_{k=1}^{K}\mathbf{x}_{np}^{(k)} - \frac{1}{K}\sum_{k=1}^{K}\mathbf{e}_{(n-s)p}^{(k)} = \mathbf{x}_{np} - \frac{1}{K}\sum_{k=1}^{K}\sum_{i=0}^{s-1}\Delta_{(n-i)p}^{(k)} - \frac{1}{K}\sum_{k=1}^{K}\mathbf{e}_{(n-s)p}^{(k)} \\
&= \mathbf{x}_{(n-1)p} - \frac{1}{K}\sum_{k=1}^{K}\mathcal{C}(\Delta_{(n-s)p}^{(k)}) - \frac{1}{K}\sum_{k=1}^{K}\sum_{i=0}^{s-1}\Delta_{(n-i)p}^{(k)} - \frac{1}{K}\sum_{k=1}^{K}\mathbf{e}_{(n-s)p}^{(k)} \\
&= \frac{1}{K}\sum_{k=1}^{K}(\mathbf{x}_{(n-1)p}^{(k)} + \sum_{i=0}^{s-1}\Delta_{(n-1-i)p}^{(k)}) - \frac{1}{K}\sum_{k=1}^{K}\mathcal{C}(\Delta_{(n-s)p}^{(k)}) - \frac{1}{K}\sum_{k=1}^{K}\sum_{i=0}^{s-1}\Delta_{(n-i)p}^{(k)} - \frac{1}{K}\sum_{k=1}^{K}\mathbf{e}_{(n-s)p}^{(k)} \\
&= \frac{1}{K}\sum_{k=1}^{K}(\mathbf{x}_{(n-1)p}^{(k)} + \sum_{i=0}^{s-1}\Delta_{(n-1-i)p}^{(k)}) - \frac{1}{K}\sum_{k=1}^{K}\Delta_{(n-s)p}^{(k)} - \frac{1}{K}\sum_{k=1}^{K}\sum_{i=0}^{s-1}\Delta_{(n-i)p}^{(k)} \\
&= \frac{1}{K}\sum_{k=1}^{K}\mathbf{x}_{(n-1)p}^{(k)} - \frac{1}{K}\sum_{k=1}^{K}\Delta_{np}^{(k)} \\
&= \frac{1}{K}\sum_{k=1}^{K}\mathbf{x}_{(n-1)p}^{(k)} - \frac{1}{K}\sum_{k=1}^{K}\sum_{\tau=(n-1)p}^{np-1}\eta_{\tau}\mathbf{m}_{\tau+1}^{(k)} - \frac{1}{K}\sum_{k=1}^{K}\mathbf{e}_{np-1-sp}^{(k)} \\
&= \tilde{\mathbf{x}}_{np-1} - \frac{1}{K}\sum_{k=1}^{K}\eta_{np-1}\mathbf{m}_{np}^{(k)}.
\end{aligned}
\tag{21}
$$

For $t \neq np$,

$$
\tilde{\mathbf{x}}_t = \frac{1}{K}\sum_{k=1}^{K}\mathbf{x}_t^{(k)} - \frac{1}{K}\sum_{k=1}^{K}\mathbf{e}_{t-sp}^{(k)} = \frac{1}{K}\sum_{k=1}^{K}\mathbf{x}_t^{(k)} - \frac{1}{K}\sum_{k=1}^{K}\mathbf{e}_{t-1-sp}^{(k)} = \tilde{\mathbf{x}}_{t-1} - \frac{\eta_{t-1}}{K}\sum_{k=1}^{K}\mathbf{m}_t^{(k)}.
\tag{22}
$$

$\square$

## D  PROOF OF THEOREM 1

**Lemma 4.** *For OLCO$_3$-VQ with vanilla SGD and under Assumptions 2, 3, 4, and 5, the local error satisfies*

$$
\mathbb{E}\|\boldsymbol{e}_t^{(k)}\|_2^2 \leq \frac{12(1-\delta)}{\delta^2}p^2\eta^2(\sigma^2 + \kappa^2 + G^2).
\tag{23}
$$

*Proof.* First we have

$$
\begin{aligned}
\mathbb{E}\|\nabla F(\mathbf{x}_t^{(k)};\xi_t^{(k)})\|_2^2 &\leq 3\mathbb{E}\|\nabla F(\mathbf{x}_t^{(k)};\xi_t^{(k)}) - \nabla f_k(\mathbf{x}_t^{(k)})\|^2 + 3\mathbb{E}\|\nabla f_k(\mathbf{x}_t^{(k)}) - \nabla f(\mathbf{x}_t^{(k)})\|_2^2 + 3\mathbb{E}\|\nabla f(\mathbf{x}_t^{(k)})\|_2^2 \\
&\leq 3\sigma^2 + 3\kappa^2 + 3G^2.
\end{aligned}
\tag{24}
$$

Let $S_t = \lfloor\frac{t}{p}\rfloor$,

$$
\begin{aligned}
\mathbb{E}\|\mathbf{e}_t^{(k)}\|_2^2 &= \mathbb{E}\|\mathbf{e}_{S_t p}^{(k)}\|_2^2 = \mathbb{E}\|\mathcal{C}(\Delta_{S_t p}^{(k)}) - \Delta_{S_t p}^{(k)}\|_2^2 \leq (1-\delta)\mathbb{E}\|\Delta_{S_t p}^{(k)}\|_2^2 \\
&= (1-\delta)\mathbb{E}\|\sum_{t'=(S_t-1)p}^{S_t p - 1}\eta\nabla F(\mathbf{x}_{t'}^{(k)};\xi_{t'}^{(k)}) + \mathbf{e}_{(S_t-s-1)p}^{(k)}\|_2^2 \\
&\leq (1-\delta)(1+\rho)\mathbb{E}\|\mathbf{e}_{(S_t-s-1)p}^{(k)}\|_2^2 + (1+\delta)(1+\frac{1}{\rho})\mathbb{E}\|\sum_{t'=(S_t-1)p}^{S_t p - 1}\eta\nabla F(\mathbf{x}_{t'}^{(k)};\xi_{t'}^{(k)})\|_2^2 \\
&\leq (1-\delta)(1+\rho)\mathbb{E}\|\mathbf{e}_{(S_t-s-1)p}^{(k)}\|_2^2 + 3(1+\delta)(1+\frac{1}{\rho})p^2\eta^2(\sigma^2 + \kappa^2 + G^2).
\end{aligned}
\tag{25}
$$

Therefore,

$$
\mathbb{E}\|\mathbf{e}_t^{(k)}\|_2^2 \le 3(1-\delta)(1+\frac{1}{\rho})p^2\eta^2(\sigma^2+\kappa^2+G^2)\sum_{i=0}^{\lfloor\frac{S_t}{s}\rfloor-1}[(1-\delta)(1+\rho)]^i
$$

$$
\le \frac{3(1-\delta)(1+\frac{1}{\rho})}{1-(1-\delta)(1+\rho)}p^2\eta^2(\sigma^2+\kappa^2+G^2)\,. \tag{26}
$$

Let $\rho = \frac{\delta}{2(1-\delta)}$ such that $1+\frac{1}{\rho} = \frac{2-\delta}{\delta} \le \frac{2}{\delta}$, then $\mathbb{E}\|\mathbf{e}_t^{(k)}\|_2^2 \le \frac{12(1-\delta)}{\delta^2}p^2\eta^2(\sigma^2+\kappa^2+G^2)$. $\qquad\square$

**Lemma 5.** *For OLCO$_3$-VQ with vanilla SGD and under Assumptions 1, 2, 3, 4, and 5, if the learning rate $\eta \le \frac{1}{6L(s+1)p}$, we have*

$$
\frac{1}{KT}\sum_{t=0}^{T-1}\sum_{k=1}^{K-1}\mathbb{E}\|\tilde{\boldsymbol{x}}_t - \boldsymbol{x}_t^{(k)}\|_2^2
$$

$$
\le 3\eta^2(s+1)p\sigma^2(1+\frac{12(1-\delta)}{\delta^2}(s+1)p) + 9\eta^2(s+1)^2p^2\kappa^2(1+\frac{4(1-\delta)}{\delta^2}) \tag{27}
$$

$$
+ \frac{36(1-\delta)}{\delta^2}\eta^2(s+1)^2p^2G^2\,.
$$

*Proof.* Let $S_t = \lfloor\frac{t}{p}\rfloor$,

$$
\frac{1}{K}\sum_{k=1}^{K}\mathbb{E}\|\tilde{\mathbf{x}}_t - \mathbf{x}_t^{(k)}\|_2^2 = \frac{1}{K}\sum_{k=1}^{K}\mathbb{E}\|\frac{1}{K}\sum_{k'=1}^{K}(-\sum_{i=0}^{s-1}\Delta_{(S_t-i)p}^{(k')} - \sum_{t'=S_tp}^{t-1}\eta\nabla F_{k'}(\mathbf{x}_{t'}^{(k')};\xi_{t'}^{(k')}))
$$

$$
-(-\sum_{i=0}^{s-1}\Delta_{(S_t-i)p}^{(k)} - \sum_{t'=S_tp}^{t-1}\eta\nabla F_k(\mathbf{x}_{t'}^{(k)};\xi_{t'}^{(k)})) - \frac{1}{K}\sum_{k'=1}^{K}\mathbf{e}_{t-sp}^{(k')}\|_2^2
$$

$$
= \frac{1}{K}\sum_{k=1}^{K}\mathbb{E}\| - \frac{1}{K}\sum_{k'=1}^{K}\sum_{t'=(S_t-s)p}^{t-1}\eta\nabla F_{k'}(\mathbf{x}_{t'}^{(k')};\xi_{t'}^{(k')}) + \sum_{t'=(S_t-s)p}^{t-1}\eta\nabla F_k(\mathbf{x}_{t'}^{(k)};\xi_{t'}^{(k)})
$$

$$
- \frac{1}{K}\sum_{k'=1}^{K}\mathbf{e}_{t-sp}^{(k')} - \frac{1}{K}\sum_{k'=1}^{K}\sum_{i=0}^{s-1}\mathbf{e}_{(S_t-i-s-1)p}^{(k')} + \sum_{i=0}^{s-1}\mathbf{e}_{(S_t-i-s-1)p}^{(k)}\|_2^2
$$

$$
\le \frac{2\eta^2}{K}\sum_{k=1}^{K}\mathbb{E}\| - \frac{1}{K}\sum_{k'=1}^{K}\sum_{t'=(S_t-s)p}^{t-1}\nabla F_{k'}(\mathbf{x}_{t'}^{(k')};\xi_{t'}^{(k')}) + \sum_{t'=(S_t-s)p}^{t-1}\nabla F_k(\mathbf{x}_{t'}^{(k)};\xi_{t'}^{(k)})\|_2^2
$$

$$
+ \frac{2}{K}\sum_{k=1}^{K}\mathbb{E}\| - \frac{1}{K}\sum_{k'=1}^{K}\mathbf{e}_{t-sp}^{(k')} - \frac{1}{K}\sum_{k'=1}^{K}\sum_{i=0}^{s-1}\mathbf{e}_{(S_t-i-s-1)p}^{(k')} + \sum_{i=0}^{s-1}\mathbf{e}_{(S_t-i-s-1)p}^{(k)}\|_2^2\,. \tag{28}
$$

The first term is bounded by

$$
\frac{2\eta^2}{K} \sum_{k=1}^{K} \mathbb{E} \| - \frac{1}{K} \sum_{k'=1}^{K} \sum_{t'=(S_t-s)p}^{t-1} \nabla F_{k'}(\mathbf{x}_{t'}^{(k')}; \xi_{t'}^{(k')}) + \sum_{t'=(S_t-s)p}^{t-1} \nabla F_k(\mathbf{x}_{t'}^{(k)}; \xi_{t'}^{(k)}) \|_2^2
$$

$$
\leq \frac{2\eta^2}{K} \sum_{k=1}^{K} \mathbb{E} \left\| \sum_{t'=(S_t-s)p}^{t-1} \left( -\frac{1}{K} \sum_{k'=1}^{K} (\nabla F_{k'}(\mathbf{x}_{t'}^{(k')}; \xi_{t'}^{(k')}) - \nabla f_{k'}(\mathbf{x}_{t'}^{(k')})) + (\nabla F_k(\mathbf{x}_{t'}^{(k)}; \xi_{t'}^{(k)}) - \nabla f_k(\mathbf{x}_{t'}^{(k)})) \right) \right\|_2^2
$$

$$
+ \frac{2\eta^2}{K} \sum_{k=1}^{K} \mathbb{E} \| \sum_{t'=(S_t-s)p}^{t-1} (-\frac{1}{K} \sum_{k'=1}^{K} \nabla f_{k'}(\mathbf{x}_{t'}^{(k')}) + \nabla f_k(\mathbf{x}_{t'}^{(k)})) \|_2^2
$$

$$
= \frac{2\eta^2}{K} \sum_{k=1}^{K} \sum_{t'=(S_t-s)p}^{t-1} \mathbb{E} \| -\frac{1}{K} \sum_{k'=1}^{K} (\nabla F_{k'}(\mathbf{x}_{t'}^{(k')}; \xi_{t'}^{(k')}) - \nabla f_{k'}(\mathbf{x}_{t'}^{(k')})) + (\nabla F_k(\mathbf{x}_{t'}^{(k)}; \xi_{t'}^{(k)}) - \nabla f_k(\mathbf{x}_{t'}^{(k)})) \|_2^2
$$

$$
+ \frac{2\eta^2}{K} \sum_{k=1}^{K} \mathbb{E} \| \sum_{t'=(S_t-s)p}^{t-1} (-\frac{1}{K} \sum_{k'=1}^{K} \nabla f_{k'}(\mathbf{x}_{t'}^{(k')}) + \nabla f_k(\mathbf{x}_{t'}^{(k)})) \|_2^2
$$

$$
\leq \frac{2\eta^2}{K} \sum_{k=1}^{K} \sum_{t'=(S_t-s)p}^{t-1} \mathbb{E} \| \nabla F_k(\mathbf{x}_{t'}^{(k)}; \xi_{t'}^{(k)}) - \nabla f_k(\mathbf{x}_{t'}^{(k)}) \|_2^2
$$

$$
+ \frac{2\eta^2}{K} \sum_{k=1}^{K} \sum_{t'=(S_t-s)p}^{t-1} (t - (S_t-s)p) \mathbb{E} \| -\frac{1}{K} \sum_{k'=1}^{K} \nabla f_{k'}(\mathbf{x}_{t'}^{(k')}) + \nabla f_k(\mathbf{x}_{t'}^{(k)}) \|_2^2
$$

$$
\leq 2\eta^2(s+1)p\sigma^2 + \frac{2\eta^2(s+1)p}{K} \sum_{t'=t-(s+1)p}^{t-1} \sum_{k=1}^{K} \mathbb{E} \| -\frac{1}{K} \sum_{k'=1}^{K} \nabla f_{k'}(\mathbf{x}_{t'}^{(k')}) + \nabla f_k(\mathbf{x}_{t'}^{(k)}) \|_2^2,
$$

$$
\tag{29}
$$

where the third inequality follows $\frac{1}{K} \sum_{k=1}^{K} \| \frac{1}{K} \sum_{k'=1}^{K} a_{k'} - a_k \|_2^2 = \frac{1}{K} \sum_{k=1}^{K} \| a_k \|_2^2 - \| \frac{1}{K} \sum_{k=1}^{K} a_k \|_2^2 \leq \frac{1}{K} \sum_{k=1}^{K} \| a_k \|_2^2$, and

$$
\frac{1}{K} \sum_{k=1}^{K} \mathbb{E} \| -\frac{1}{K} \sum_{k'=1}^{K} \nabla f_{k'}(\mathbf{x}_t^{k'}) + \nabla f_k(\mathbf{x}_t^{(k)}) \|_2^2
$$

$$
= \frac{3}{K} \sum_{k=1}^{K} \mathbb{E} [\| \nabla f_k(\mathbf{x}_t^{(k)}) - \nabla f_k(\tilde{\mathbf{x}}_t) \|_2^2 + \| \nabla f_k(\tilde{\mathbf{x}}_t) - \nabla f(\tilde{\mathbf{x}}_t) \|_2^2 + \| \nabla f(\tilde{\mathbf{x}}_t) - \frac{1}{K} \sum_{k'=1}^{K} \nabla f_{k'}(\mathbf{x}_t^{k'}) \|_2^2]
$$

$$
\leq \frac{3}{K} \sum_{k=1}^{K} \mathbb{E} [L^2 \| \tilde{\mathbf{x}}_t - \mathbf{x}_t^{(k)} \|_2^2 + \kappa^2 + \frac{1}{K} \sum_{k'=1}^{K} \| \nabla f_{k'}(\tilde{\mathbf{x}}_t) - \nabla f_{k'}(\mathbf{x}_t^{k'}) \|_2^2]
$$

$$
\leq \frac{6L^2}{K} \sum_{k=1}^{K} \mathbb{E} \| \tilde{\mathbf{x}}_t - \mathbf{x}_t^{(k)} \|_2^2 + 3\kappa^2 .
$$

$$
\tag{30}
$$

The second term is bounded by

$$
\frac{2}{K}\sum_{k=1}^{K}\mathbb{E}\|-\frac{1}{K}\sum_{k'=1}^{K}\mathbf{e}_{t-sp}^{(k')}-\frac{1}{K}\sum_{k'=1}^{K}\sum_{i=0}^{s-1}\mathbf{e}_{(S_t-i-s-1)p}^{(k')}+\sum_{i=0}^{s-1}\mathbf{e}_{(S_t-i-s-1)p}^{(k)}\|_2^2
$$

$$
\leq \frac{2(1+s)}{K}\sum_{k=1}^{K}\mathbb{E}\|\frac{1}{K}\sum_{k'=1}^{K}\mathbf{e}_{(S_t-s)p}^{(k')}\|_2^2+\frac{2(1+\frac{1}{s})}{K}\sum_{k=1}^{K}\mathbb{E}\|-\frac{1}{K}\sum_{k'=1}^{K}\sum_{i=0}^{s-1}\mathbf{e}_{(S_t-i-s-1)p}^{(k')}+\sum_{i=0}^{s-1}\mathbf{e}_{(S_t-i-s-1)p}^{(k)}\|_2^2
$$

$$
\leq \frac{2(1+s)}{K}\sum_{k=1}^{K}\mathbb{E}\|\mathbf{e}_{(S_t-s)p}^{(k)}\|_2^2+\frac{2(1+\frac{1}{s})}{K}\sum_{k=1}^{K}\mathbb{E}\|\sum_{i=0}^{s-1}\mathbf{e}_{(S_t-i-s-1)p}^{(k)}\|_2^2
$$

$$
\leq \frac{2(1+s)}{K}\sum_{k=1}^{K}\mathbb{E}\|\mathbf{e}_{(S_t-s)p}^{(k)}\|_2^2+\frac{2(1+s)}{K}\sum_{k=1}^{K}\sum_{i=0}^{s-1}\mathbb{E}\|\mathbf{e}_{(S_t-i-s-1)p}^{(k)}\|_2^2
$$

$$
= \frac{2(s+1)}{K}\sum_{k=1}^{K}\sum_{i=0}^{s}\mathbb{E}\|\mathbf{e}_{(S_t-i-s)p}^{(k)}\|_2^2\leq \frac{24(1-\delta)}{\delta^2}(s+1)^2p^2\eta^2(\sigma^2+\kappa^2+G^2),
$$

$$(31)$$

where the last inequality follows Lemma 4. Combine the bounds of the first term and the second term, we have

$$
\frac{1}{K}\sum_{k=1}^{K}\mathbb{E}\|\tilde{\mathbf{x}}_t-\mathbf{x}_t^{(k)}\|_2^2
$$

$$
\leq 2\eta^2(s+1)p\sigma^2+2\eta^2(s+1)p\sum_{t'=t-(s+1)p}^{t-1}\left(\frac{6L^2}{K}\sum_{k=1}^{K}\mathbb{E}\|\tilde{\mathbf{x}}_t-\mathbf{x}_t^{(k)}\|_2^2+3\kappa^2\right)
$$

$$
+\frac{24(1-\delta)}{\delta^2}(s+1)^2p^2\eta^2(\sigma^2+\kappa^2+G^2)
$$

$$
\leq 2\eta^2(s+1)p\sigma^2(1+\frac{12(1-\delta)}{\delta^2}(s+1)p)+6\eta^2(s+1)^2p^2\kappa^2(1+\frac{4(1-\delta)}{\delta^2})+\frac{24(1-\delta)}{\delta^2}\eta^2(s+1)^2p^2G^2
$$

$$
+12\eta^2L^2(s+1)p\sum_{t'=t-(s+1)p}^{t-1}\frac{1}{K}\sum_{k=1}^{K}\mathbb{E}\|\tilde{\mathbf{x}}_t-\mathbf{x}_t^{(k)}\|_2^2.
$$

$$(32)$$

Sum the above inequality from $t=0$ to $t=T-1$ and divide it by $T$,

$$
\frac{1}{KT}\sum_{t=0}^{T-1}\sum_{k=1}^{K-1}\mathbb{E}\|\tilde{\mathbf{x}}_t-\mathbf{x}_t^{(k)}\|_2^2
$$

$$
\leq 2\eta^2(s+1)p\sigma^2(1+\frac{12(1-\delta)}{\delta^2}(s+1)p)+6\eta^2(s+1)^2p^2\kappa^2(1+\frac{4(1-\delta)}{\delta^2})+\frac{24(1-\delta)}{\delta^2}\eta^2(s+1)^2p^2G^2
$$

$$
+12\eta^2L^2(s+1)^2p^2\cdot\frac{1}{KT}\sum_{t=0}^{T-1}\sum_{k=1}^{K-1}\mathbb{E}\|\tilde{\mathbf{x}}_t-\mathbf{x}_t^{(k)}\|_2^2.
$$

$$(33)$$

Therefore,

$$
\frac{1}{KT}\sum_{t=0}^{T-1}\sum_{k=1}^{K-1}\mathbb{E}\|\tilde{\mathbf{x}}_t-\mathbf{x}_t^{(k)}\|_2^2
$$

$$
\leq \frac{2\eta^2(s+1)p\sigma^2(1+\frac{12(1-\delta)}{\delta^2}(s+1)p)+6\eta^2(s+1)^2p^2\kappa^2(1+\frac{4(1-\delta)}{\delta^2})+\frac{24(1-\delta)}{\delta^2}\eta^2(s+1)^2p^2G^2}{1-12\eta^2L^2(s+1)^2p^2}.
$$

$$(34)$$

If we choose $\eta \leq \frac{1}{6L(s+1)p}$,

$$\frac{1}{KT} \sum_{t=0}^{T-1} \sum_{k=1}^{K-1} \mathbb{E}\|\tilde{\mathbf{x}}_t - \mathbf{x}_t^{(k)}\|_2^2$$

$$\leq 3\eta^2(s+1)p\sigma^2(1 + \frac{12(1-\delta)}{\delta^2}(s+1)p) + 9\eta^2(s+1)^2 p^2 \kappa^2(1 + \frac{4(1-\delta)}{\delta^2}) + \frac{36(1-\delta)}{\delta^2}\eta^2(s+1)^2 p^2 G^2.$$

(35)

$\square$

**Theorem 1.** *For OLCO$_3$-VQ with vanilla SGD and under Assumptions 1, 2, 3, 4, and 5, if the learning rate $\eta \leq \min\{\frac{1}{6L(s+1)p}, \frac{1}{9L}\}$, then*

$$\frac{1}{T} \sum_{t=0}^{T-1} \mathbb{E}\|\nabla f(\frac{1}{K} \sum_{k=1}^{K} \boldsymbol{x}_t^{(k)})\|_2^2 \leq \frac{6(f(\boldsymbol{x}_0) - f_*)}{\eta T} + \frac{9\eta L\sigma^2}{K} + 12\eta^2 L^2(s+1)p\sigma^2(1 + \frac{14(1-\delta)}{\delta^2}(s+1)p)$$

$$+ 36\eta^2 L^2(s+1)^2 p^2 \kappa^2(1 + \frac{5(1-\delta)}{\delta^2}) + \frac{168(1-\delta)}{\delta^2}\eta^2 L^2(s+1)^2 p^2 G^2.$$

(36)

*Proof.* According to Assumption 1,

$$\mathbb{E}_t f(\tilde{\mathbf{x}}_{t+1}) - f(\tilde{\mathbf{x}}_t) \leq \mathbb{E}_t \langle \nabla f(\tilde{\mathbf{x}}_t), \tilde{\mathbf{x}}_{t+1} - \tilde{\mathbf{x}}_t \rangle + \frac{L}{2}\mathbb{E}_t\|\tilde{\mathbf{x}}_{t+1} - \tilde{\mathbf{x}}_t\|_2^2$$

$$= -\eta \langle \nabla f(\tilde{\mathbf{x}}_t), \frac{1}{K} \sum_{k=1}^{K} \nabla f_k(\mathbf{x}_t^{(k)}) \rangle + \frac{L\eta^2}{2}\mathbb{E}_t\|\frac{1}{K} \sum_{k=1}^{K} \nabla F_k(\mathbf{x}_t^{(k)}; \xi_t^{(k)})\|_2^2.$$

(37)

For the first term,

$$-\langle \nabla f(\tilde{\mathbf{x}}_t), \frac{1}{K} \sum_{k=1}^{K} \nabla f_k(\mathbf{x}_t^{(k)}) \rangle = -\|\nabla f(\tilde{\mathbf{x}}_t)\|_2^2 - \langle \nabla f(\tilde{\mathbf{x}}_t), \frac{1}{K} \sum_{k=1}^{K} (\nabla f_k(\mathbf{x}_t^{(k)}) - \nabla f_k(\tilde{\mathbf{x}}_t)) \rangle$$

$$\leq -\frac{1}{2}\|\nabla f(\tilde{\mathbf{x}}_t)\|_2^2 + \frac{1}{2}\|\frac{1}{K} \sum_{k=1}^{K} (\nabla f_k(\mathbf{x}_t^{(k)}) - \nabla f_k(\tilde{\mathbf{x}}_t))\|_2^2$$

(38)

$$\leq -\frac{1}{2}\|\nabla f(\tilde{\mathbf{x}}_t)\|_2^2 + \frac{L^2}{2K} \sum_{k=1}^{K} \|\tilde{\mathbf{x}}_t - \mathbf{x}_t^{(k)}\|_2^2,$$

where the first equality follows that $\nabla f(\tilde{\mathbf{x}}_t) = \frac{1}{K} \sum_{k=1}^{K} \nabla f_k(\tilde{\mathbf{x}}_t)$. For the second term,

$$\mathbb{E}_t\|\frac{1}{K} \sum_{k=1}^{K} \nabla F_k(\mathbf{x}_t^{(k)}; \xi_t^{(k)})\|_2^2$$

$$= \mathbb{E}_t\|\frac{1}{K} \sum_{k=1}^{K} (\nabla F_k(\mathbf{x}_t^{(k)}; \xi_t^{(k)}) - \nabla f_k(\mathbf{x}_t^{(k)})) + \frac{1}{K} \sum_{k=1}^{K} (\nabla f_k(\mathbf{x}_t^{(k)}) - \nabla f_k(\tilde{\mathbf{x}}_t)) + \nabla f(\tilde{\mathbf{x}}_t)\|_2^2$$

$$\leq 3\mathbb{E}_t\|\frac{1}{K} \sum_{k=1}^{K} (\nabla F_k(\mathbf{x}_t^{(k)}; \xi_t^{(k)}) - \nabla f_k(\mathbf{x}_t^{(k)}))\|_2^2 + 3\|\frac{1}{K} \sum_{k=1}^{K} (\nabla f_k(\mathbf{x}_t^{(k)}) - \nabla f_k(\tilde{\mathbf{x}}_t))\|_2^2 + 3\|\nabla f(\tilde{\mathbf{x}}_t)\|_2^2$$

$$\leq \frac{3\sigma^2}{K} + \frac{3L^2}{K} \sum_{k=1}^{K} \|\tilde{\mathbf{x}}_t - \mathbf{x}_t^{(k)}\|_2^2 + 3\|\nabla f(\tilde{\mathbf{x}}_t)\|_2^2.$$

(39)

Combine them and we have

$$\mathbb{E}_t(\tilde{\mathbf{x}}_{t+1}) - f(\tilde{\mathbf{x}}_t) \leq -\frac{\eta}{2}(1 - 3\eta L)\|\nabla f(\tilde{\mathbf{x}}_t)\|_2^2 + \frac{\eta L^2}{2}(1 + 3\eta L)\frac{1}{K} \sum_{k=1}^{K} \|\tilde{\mathbf{x}}_t - \mathbf{x}_t^{(k)}\|_2^2 + \frac{3\eta^2 L\sigma^2}{2K}.$$

(40)

If we choose $\eta \leq \frac{1}{9L}$,

$$\mathbb{E}_t(\tilde{\mathbf{x}}_{t+1}) - f(\tilde{\mathbf{x}}_t) \leq -\frac{\eta}{3}\|\nabla f(\tilde{\mathbf{x}}_t)\|_2^2 + \frac{2\eta L^2}{3}\frac{1}{K}\sum_{k=1}^{K}\|\tilde{\mathbf{x}}_t - \mathbf{x}_t^{(k)}\|_2^2 + \frac{3\eta^2 L\sigma^2}{2K}. \qquad (41)$$

Then for the averaged parameters $\frac{1}{K}\sum_{k=1}^{K}\mathbf{x}_t^{(k)}$,

$$\|\nabla f(\frac{1}{K}\sum_{k=1}^{K}\mathbf{x}_t^{(k)})\|_2^2 \leq 2\|\nabla f(\frac{1}{K}\sum_{k=1}^{K}\mathbf{x}_t^{(k)}) - \nabla f(\tilde{\mathbf{x}}_t)\|_2^2 + 2\|\nabla f(\tilde{\mathbf{x}}_t)\|_2^2$$

$$\leq 2L^2\|\frac{1}{K}\sum_{k=1}^{K}\mathbf{e}_{t-sp}^{(k)}\|_2^2 + 2\|\nabla f(\tilde{\mathbf{x}}_t)\|_2^2 \leq \frac{2L^2}{K}\sum_{k=1}^{K}\|\mathbf{e}_{t-sp}^{(k)}\|_2^2 + 2\|\nabla f(\tilde{\mathbf{x}}_t)\|_2^2$$

$$\leq \frac{6[f(\tilde{\mathbf{x}}_t) - \mathbb{E}_t f(\tilde{\mathbf{x}}_{t+1})]}{\eta} + \frac{9\eta L\sigma^2}{K} + \frac{4L^2}{K}\sum_{k=1}^{K}\|\tilde{\mathbf{x}}_t - \mathbf{x}_t^{(k)}\|_2^2 + \frac{2L^2}{K}\sum_{k=1}^{K}\|\mathbf{e}_{t-sp}^{(k)}\|_2^2. \tag{42}$$

Take total expectation, sum from $t = 0$ to $t = T - 1$, and rearrange,

$$\frac{1}{T}\sum_{t=0}^{T-1}\mathbb{E}\|\nabla f(\frac{1}{K}\sum_{k=1}^{K}\mathbf{x}_t^{(k)})\|_2^2$$

$$\leq \frac{6[f(\tilde{\mathbf{x}}_0) - \mathbb{E}f(\tilde{\mathbf{x}_T})]}{\eta T} + \frac{9\eta L\sigma^2}{K} + \frac{2L^2}{KT}\sum_{t=0}^{T-1}\sum_{k=1}^{K}\mathbb{E}\|\mathbf{e}_{t-sp}^{(k)}\|_2^2 + \frac{4L^2}{KT}\sum_{t=0}^{T-1}\sum_{k=1}^{K}\mathbb{E}\|\tilde{\mathbf{x}}_t - \mathbf{x}_t^{(k)}\|_2^2$$

$$\leq \frac{6[f(\tilde{\mathbf{x}}_0) - \mathbb{E}f(\tilde{\mathbf{x}_T})]}{\eta T} + \frac{9\eta L\sigma^2}{K} + \frac{24(1-\delta)}{\delta^2}p^2\eta^2 L^2(\sigma^2 + \kappa^2 + G^2)$$

$$+ 12\eta^2 L^2(s+1)p\sigma^2(1 + \frac{12(1-\delta)}{\delta^2}(s+1)p) + 36\eta^2 L^2(s+1)^2 p^2\kappa^2(1 + \frac{4(1-\delta)}{\delta^2}) \tag{43}$$

$$+ \frac{144(1-\delta)}{\delta^2}\eta^2 L^2(s+1)^2 p^2 G^2$$

$$\leq \frac{6(f(\mathbf{x}_0) - f_*)}{\eta T} + \frac{9\eta L\sigma^2}{K} + 12\eta^2 L^2(s+1)p\sigma^2(1 + \frac{14(1-\delta)}{\delta^2}(s+1)p)$$

$$+ 36\eta^2 L^2(s+1)^2 p^2\kappa^2(1 + \frac{5(1-\delta)}{\delta^2}) + \frac{168(1-\delta)}{\delta^2}\eta^2 L^2(s+1)^2 p^2 G^2,$$

where the second inequality follows Lemma 4 and 5. $\qquad\qquad \square$

## E    PROOF OF THEOREM 2

**Lemma 6.** *For OLCO$_3$-TC with vanilla SGD and under Assumptions 2, 3, 4, and 5, the local error satisfies*

$$\mathbb{E}\|\mathbf{e}_t^{(k)}\|_2^2 \leq \frac{12(1-\delta)}{\delta^2}p^2\eta^2(\sigma^2 + \kappa^2 + G^2). \tag{44}$$

*Proof.* Same as the proof of Lemma 4, except that $\mathbf{e}_{(S_t-s-1)p}^{(k)}$ is replaced with $\mathbf{e}_{(S_t-1)p}^{(k)}$. $\qquad \square$

**Lemma 7.** *For OLCO$_3$-TC with vanilla SGD and under Assumptions 2, 3, 4, and 5, the server error satisfies*

$$\mathbb{E}\|\mathbf{e}_t\|_2^2 \leq \frac{96(2-\delta)(1-\delta)}{\delta^4}p^2\eta^2(\sigma^2 + \kappa^2 + G^2). \tag{45}$$

*Proof.* Let $S_t = \lfloor \frac{t}{p} \rfloor$,

$$\mathbb{E}\|\frac{1}{K}\sum_{k=1}^{K}\mathcal{C}(\Delta_{S_tp}^{(k)})\|_2^2 \leq 2\mathbb{E}\|\frac{1}{K}\sum_{k=1}^{K}\mathcal{C}(\Delta_{S_tp}^{(k)}) - \frac{1}{K}\sum_{k=1}^{K}\Delta_{S_tp}^{(k)}\|_2^2 + 2\mathbb{E}\|\frac{1}{K}\sum_{k=1}^{K}\Delta_{S_tp}^{(k)}\|_2^2$$

$$\leq \frac{2}{K}\sum_{k=1}^{K}\mathbb{E}\|\mathcal{C}(\Delta_{S_tp}^{(k)}) - \Delta_{S_tp}^{(k)}\|_2^2 + \frac{2}{K}\sum_{k=1}^{K}\mathbb{E}\|\Delta_{S_tp}^{(k)}\|_2^2 \quad (46)$$

$$\leq \frac{2(2-\delta)}{K}\sum_{k=1}^{K}\mathbb{E}\|\Delta_{S_tp}^{(k)}\|_2^2 .$$

Following the proof of Lemma 4 we have $\mathbb{E}\|\Delta_{S_tp}^{(k)}\|_2^2 \leq \frac{3(1+\frac{1}{\rho})}{1-(1-\delta)(1+\rho)}p^2\eta^2(\sigma^2+\kappa^2+G^2)$. Therefore,

$$\mathbb{E}\|\mathbf{e}_t\|_2^2 = \mathbb{E}\|\mathbf{e}_{S_tp}\|_2^2 \leq (1-\delta)\mathbb{E}\|\frac{1}{K}\sum_{k=1}^{K}\mathcal{C}(\Delta_{S_tp}^{(k)}) + \mathbf{e}_{(S_t-1)p}\|^2$$

$$\leq (1-\delta)(1+\frac{1}{\rho})\mathbb{E}\|\frac{1}{K}\sum_{k=1}^{K}\mathcal{C}(\Delta_{S_tp}^{(k)})\|_2^2 + (1-\delta)(1+\rho)\mathbb{E}\|\mathbf{e}_{(S_t-1)p}\|_2^2$$

$$\leq 2(2-\delta)(1-\delta)(1+\frac{1}{\rho})\frac{1}{K}\sum_{k=1}^{K}\mathbb{E}\|\Delta_{S_tp}^{(k)}\|_2^2 + (1-\delta)(1+\rho)\mathbb{E}\|\mathbf{e}_{(S_t-1)p}\|_2^2$$

$$\leq 2(2-\delta)(1-\delta)(1+\frac{1}{\rho})\frac{3(1+\frac{1}{\rho})}{1-(1-\delta)(1+\rho)}p^2\eta^2(\sigma^2+\kappa^2+G^2) + (1-\delta)(1+\rho)\mathbb{E}\|\mathbf{e}_{(S_t-1)p}\|_2^2$$

$$\leq \frac{6(2-\delta)(1-\delta)(1+\frac{1}{\rho})^2}{[1-(1-\delta)(1+\rho)]^2}p^2\eta^2(\sigma^2+\kappa^2+G^2) .$$

$$(47)$$

Let $\rho = \frac{\delta}{2(1-\delta)}$ such that $1+\frac{1}{\rho} = \frac{2-\delta}{\delta} \leq \frac{2}{\delta}$, then $\mathbb{E}\|\mathbf{e}_t^{(k)}\|_2^2 \leq \frac{96(2-\delta)(1-\delta)}{\delta^4}p^2\eta^2(\sigma^2+\kappa^2+G^2)$. $\square$

**Lemma 8.** *For OLCO$_3$-TC with vanilla SGD and under Assumptions 1, 2, 3, 4, and 5, if the learning rate $\eta \leq \frac{1}{6L(s+1)p}$ and let $h(\delta) = \frac{1-\delta}{\delta^2}(1 + \frac{4(2-\delta)}{\delta^2})$, we have*

$$\frac{1}{KT}\sum_{t=0}^{T-1}\sum_{k=1}^{K-1}\mathbb{E}\|\tilde{\boldsymbol{x}}_t - \boldsymbol{x}_t^{(k)}\|_2^2 \leq 3\eta^2 p\sigma^2(s+1+72h(\delta)p) + 9\eta^2p^2\kappa^2((s+1)^2+24h(\delta)) + 216h(\delta)\eta^2p^2G^2 .$$

$$(48)$$

*Proof.* Let $S_t = \lfloor \frac{t}{p} \rfloor$,

$$\frac{1}{K}\sum_{k=1}^{K}\mathbb{E}\|\tilde{\boldsymbol{x}}_t - \boldsymbol{x}_t^{(k)}\|_2^2 = \frac{1}{K}\sum_{k=1}^{K}\mathbb{E}\|\frac{1}{K}\sum_{k'=1}^{K}(-\sum_{i=0}^{s-1}\mathcal{C}(\Delta_{(S_t-i)p}^{(k')}) - \sum_{t'=S_tp}^{t-1}\eta\nabla F_{k'}(\mathbf{x}_{t'}^{(k')};\xi_{t'}^{(k')}))$$

$$- (-\sum_{i=0}^{s-1}\mathcal{C}(\Delta_{(S_t-i)p}^{(k)}) - \sum_{t'=S_tp}^{t-1}\eta\nabla F_k(\mathbf{x}_{t'}^{(k)};\xi_{t'}^{(k)})) - \frac{1}{K}\sum_{k'=1}^{K}\mathbf{e}_t^{(k')} - \mathbf{e}_{t-sp}\|_2^2 ,$$

$$(49)$$

where

$$\sum_{i=0}^{s-1}\mathcal{C}(\Delta_{(S_t-i)p}^{(k)}) = \sum_{i=0}^{s-1}[\Delta_{(S_t-i)p}^{(k)} - \mathbf{e}_{(S_t-i)p}^{(k)}] = \sum_{i=0}^{s-1}[\sum_{t'=(S_t-i-1)p}^{(S_t-i)p-1}\eta\nabla F_k(\mathbf{x}_{t'}^{(k)};\xi_{t'}^{(k)}) + \mathbf{e}_{(S_t-i-1)p}^{(k)} - \mathbf{e}_{(S_t-i)p}^{(k)}]$$

$$= \sum_{i=0}^{s-1}\sum_{t'=(S_t-i-1)p}^{(S_t-i)p-1}\nabla F_k(\mathbf{x}_{t'}^{(k)};\xi_{t'}^{(k)}) + \mathbf{e}_{(S_t-s)p}^{(k)} - \mathbf{e}_{S_tp}^{(k)} .$$

$$(50)$$

Therefore

$$
\frac{1}{K} \sum_{k=1}^{K} \mathbb{E} \| \tilde{\mathbf{x}}_t - \mathbf{x}_t^{(k)} \|_2^2
$$

$$
= \frac{1}{K} \sum_{k=1}^{K} \mathbb{E} \| - \frac{1}{K} \sum_{k'=1}^{K} \sum_{t'=(S_t-s)p}^{t-1} \eta \nabla F_{k'}(\mathbf{x}_{t'}^{(k')}; \xi_{t'}^{(k')}) + \sum_{t'=(S_t-s)p}^{t-1} \eta \nabla F_k(\mathbf{x}_{t'}^{(k)}; \xi_{t'}^{(k)})
$$

$$
- \frac{1}{K} \sum_{k'=1}^{K} (\mathbf{e}_{(S_t-s)p}^{(k')} - \mathbf{e}_{S_t p}^{(k')}) + (\mathbf{e}_{(S_t-s)p}^{(k)} - \mathbf{e}_{S_t p}^{(k)}) - \frac{1}{K} \sum_{k'=1}^{K} \mathbf{e}_t^{(k')} - \mathbf{e}_{t-sp} \|_2^2 \qquad (51)
$$

$$
\leq \frac{2\eta^2}{K} \sum_{k=1}^{K} \mathbb{E} \| - \frac{1}{K} \sum_{k'=1}^{K} \sum_{t'=(S_t-s)p}^{t-1} \eta \nabla F_{k'}(\mathbf{x}_{t'}^{(k')}; \xi_{t'}^{(k')}) + \sum_{t'=(S_t-s)p}^{t-1} \eta \nabla F_k(\mathbf{x}_{t'}^{(k)}; \xi_{t'}^{(k)}) \|_2^2
$$

$$
+ \frac{2}{K} \sum_{k=1}^{K} \mathbb{E} \| - \frac{1}{K} \sum_{k'=1}^{K} \mathbf{e}_{(S_t-s)p}^{(k')} + \mathbf{e}_{(S_t-s)p}^{(k)} - \mathbf{e}_{S_t p}^{(k)} - \mathbf{e}_{(S_t-s)p} \|_2^2,
$$

where the first term can be bounded following Eqs. (29,30). The second term satisfies

$$
\frac{2}{K} \sum_{k=1}^{K} \mathbb{E} \| - \frac{1}{K} \sum_{k'=1}^{K} \mathbf{e}_{(S_t-s)p}^{(k')} + \mathbf{e}_{(S_t-s)p}^{(k)} - \mathbf{e}_{S_t p}^{(k)} - \mathbf{e}_{(S_t-s)p} \|_2^2
$$

$$
\leq \frac{6}{K} \sum_{k=1}^{K} \mathbb{E} \| - \frac{1}{K} \sum_{k'=1}^{K} \mathbf{e}_{(S_t-s)p}^{(k')} + \mathbf{e}_{(S_t-s)p}^{(k)} \|_2^2 + \frac{6}{K} \sum_{k=1}^{K} \mathbb{E} \| \mathbf{e}_{S_t p}^{(k)} \|_2^2 + \frac{6}{K} \sum_{k=1}^{K} \mathbb{E} \| \mathbf{e}_{(S_t-s)p} \|_2^2 \qquad (52)
$$

$$
\leq \frac{6}{K} \sum_{k=1}^{K} \mathbb{E} \| \mathbf{e}_{(S_t-s)p}^{(k)} \|_2^2 + \frac{6}{K} \sum_{k=1}^{K} \mathbb{E} \| \mathbf{e}_{S_t p}^{(k)} \|_2^2 + \frac{6}{K} \sum_{k=1}^{K} \mathbb{E} \| \mathbf{e}_{(S_t-s)p} \|_2^2
$$

$$
\leq \frac{1-\delta}{\delta^2} (1 + \frac{4(2-\delta)}{\delta^2}) \cdot 144 p^2 \eta^2 (\sigma^2 + \kappa^2 + G^2),
$$

where the last inequality follows Lemmas 6 and 7. Let $h(\delta) = \frac{1-\delta}{\delta^2}(1 + \frac{4(2-\delta)}{\delta^2})$. Combine the above two inequalities and we have

$$
\frac{1}{K} \sum_{k=1}^{K} \mathbb{E} \| \tilde{\mathbf{x}}_t - \mathbf{x}_t^{(k)} \|_2^2
$$

$$
\leq 2\eta^2 (s+1) p \sigma^2 + 2\eta^2 (s+1) p \sum_{t'=t-(s+1)p}^{t-1} \left( \frac{6L^2}{K} \sum_{k=1}^{K} \mathbb{E} \| \tilde{\mathbf{x}}_t - \mathbf{x}_t^{(k)} \|_2^2 + 3\kappa^2 \right) + 144 h(\delta) p^2 \eta^2 (\sigma^2 + \kappa^2 + G^2)
$$

$$
\leq 2\eta^2 p \sigma^2 (s+1 + 72 h(\delta) p) + 6\eta^2 p^2 \kappa^2 ((s+1)^2 + 24 h(\delta)) + 144 h(\delta) p^2 \eta^2 G^2
$$

$$
+ 12\eta^2 L^2 (s+1) p \sum_{t'=t-(s+1)p}^{t-1} \frac{1}{K} \sum_{k=1}^{K} \mathbb{E} \| \tilde{\mathbf{x}}_t - \mathbf{x}_t^{(k)} \|_2^2.
$$

$$
(53)
$$

Following Eqs. (33,34,35),

$$
\frac{1}{KT} \sum_{t=0}^{T-1} \sum_{k=1}^{K-1} \mathbb{E} \| \tilde{\mathbf{x}}_t - \mathbf{x}_t^{(k)} \|_2^2 \leq 3\eta^2 p \sigma^2 (s+1 + 72 h(\delta) p) + 9\eta^2 p^2 \kappa^2 ((s+1)^2 + 24 h(\delta)) + 216 h(\delta) \eta^2 p^2 G^2 .
$$

$$
(54)
$$

$\square$

**Theorem 2.** *For OLCO₃-TC with vanilla SGD and under Assumptions 1, 2, 3, 4, and 5, if the learning rate $\eta \leq \min\{\frac{1}{6L(s+1)p}, \frac{1}{9L}\}$ and let $h(\delta) = \frac{1-\delta}{\delta^2}(1 + \frac{4(2-\delta)}{\delta^2})$, then*

$$\frac{1}{T}\sum_{t=0}^{T-1}\mathbb{E}\|\nabla f(\frac{1}{K}\sum_{k=1}^{K}\boldsymbol{x}_t^{(k)})\|_2^2 \leq \frac{6(f(\boldsymbol{x}_0) - f_*)}{\eta T} + \frac{9\eta L\sigma^2}{K}$$
$$+ 12\eta^2 p\sigma^2(s+1+80h(\delta)p) + 12\eta^2 p^2\kappa^2(3(s+1)^2 + 80h(\delta)) + 960\eta^2 p^2 G^2 h(\delta).$$

(55)

*Proof.* Following the proof of Theorem 1, we have the same inequality as Eq. (41):

$$\mathbb{E}_t f(\tilde{\mathbf{x}}_{t+1}) - f(\tilde{\mathbf{x}}_t) \leq -\frac{\eta}{3}\|\nabla f(\tilde{\mathbf{x}}_t)\|_2^2 + \frac{2\eta L^2}{3}\frac{1}{K}\sum_{k=1}^{K}\|\tilde{\mathbf{x}}_t - \mathbf{x}_t^{(k)}\|_2^2 + \frac{3\eta^2 L\sigma^2}{2K}.$$

(56)

Then for the averaged parameters $\frac{1}{K}\sum_{k=1}^{K}\mathbf{x}_t^{(k)}$,

$$\|\nabla f(\frac{1}{K}\sum_{k=1}^{K}\mathbf{x}_t^{(k)})\|_2^2 \leq 2\|\nabla f(\frac{1}{K}\sum_{k=1}^{K}\mathbf{x}_t^{(k)}) - \nabla f(\tilde{\mathbf{x}}_t)\|_2^2 + 2\|\nabla f(\tilde{\mathbf{x}}_t)\|_2^2$$

$$\leq 2L^2\|\frac{1}{K}\sum_{k=1}^{K}\mathbf{e}_t^{(k)} + \mathbf{e}_{t-sp}\|_2^2 + 2\|\nabla f(\tilde{\mathbf{x}}_t)\|_2^2$$

$$\leq \frac{4L^2}{K}\sum_{k=1}^{K}\|\mathbf{e}_t^{(k)}\|_2^2 + \frac{4L^2}{K}\|\mathbf{e}_{t-sp}\|_2^2 + 2\|\nabla f(\tilde{\mathbf{x}}_t)\|_2^2$$

$$\leq \frac{6[f(\tilde{\mathbf{x}}_t) - \mathbb{E}_t f(\tilde{\mathbf{x}}_{t+1})]}{\eta} + \frac{9\eta L\sigma^2}{K} + \frac{4L^2}{K}\sum_{k=1}^{K}\|\tilde{\mathbf{x}}_t - \mathbf{x}_t^{(k)}\|_2^2 + \frac{4L^2}{K}\sum_{k=1}^{K}\|\mathbf{e}_t^{(k)}\|_2^2 + \frac{4L^2}{K}\|\mathbf{e}_{t-sp}\|_2^2.$$

(57)

Take total expectation, sum from $t = 0$ to $t = T - 1$, and rearrange,

$$\frac{1}{T}\sum_{t=0}^{T-1}\mathbb{E}\|\nabla f(\frac{1}{K}\sum_{k=1}^{K}\mathbf{x}_t^{(k)})\|_2^2$$

$$\leq \frac{6[f(\tilde{\mathbf{x}}_0) - \mathbb{E}f(\tilde{\mathbf{x}_T})]}{\eta T} + \frac{9\eta L\sigma^2}{K} + \frac{4L^2}{KT}\sum_{t=0}^{T-1}\sum_{k=1}^{K}\mathbb{E}\|\mathbf{e}_{t-sp}^{(k)}\|_2^2 + \frac{4L^2}{KT}\sum_{t=0}^{T-1}\mathbb{E}\|\mathbf{e}_{t-sp}\|_2^2$$

$$+ \frac{4L^2}{KT}\sum_{t=0}^{T-1}\sum_{k=1}^{K}\mathbb{E}\|\tilde{\mathbf{x}}_t - \mathbf{x}_t^{(k)}\|_2^2$$

$$\leq \frac{6[f(\tilde{\mathbf{x}}_0) - \mathbb{E}f(\tilde{\mathbf{x}_T})]}{\eta T} + \frac{9\eta L\sigma^2}{K} + 12\eta^2 p\sigma^2(s+1+72h(\delta)p + \frac{4(1-\delta)}{\delta^2}p + \frac{32(2-\delta)(1-\delta)}{\delta^4}p)$$

$$+ 12\eta^2 p^2\kappa^2(3(s+1)^2 + 72h(\delta) + \frac{4(1-\delta)}{\delta^2} + \frac{32(2-\delta)(1-\delta)}{\delta^4})$$

$$+ 12\eta^2 p^2 G^2(72h(\delta) + \frac{4(1-\delta)}{\delta^2} + \frac{32(2-\delta)(1-\delta)}{\delta^4})$$

$$\leq \frac{6(f(\tilde{\mathbf{x}}_0) - f_*)}{\eta T} + \frac{9\eta L\sigma^2}{K}$$

$$+ 12\eta^2 p\sigma^2(s+1+80h(\delta)p) + 12\eta^2 p^2\kappa^2(3(s+1)^2 + 80h(\delta)) + 960\eta^2 p^2 G^2 h(\delta),$$

(58)

where the second inequality follows Lemmas 6, 7 and 8. □

## F   PROOF OF THEOREM 3

We first define two virtual variables $\mathbf{z}_t$ and $\mathbf{p}_t$ satisfying

$$\mathbf{p}_t = \begin{cases} \frac{\mu}{1-\mu}(\tilde{\mathbf{x}}_t - \tilde{\mathbf{x}}_{t-1}), & t \geq 1 \\ \mathbf{0}, & t = 0 \end{cases}$$

(59)

and

$$\mathbf{z}_t = \tilde{\mathbf{x}}_t + \mathbf{p}_t. \tag{60}$$

Then the update rule of $\mathbf{z}_t$ satisfies

$$
\begin{aligned}
\mathbf{z}_{t+1} - \mathbf{z}_t &= (\tilde{\mathbf{x}}_{t+1} - \tilde{\mathbf{x}}_t) + \frac{\mu}{1-\mu}(\tilde{\mathbf{x}}_{t+1} - \tilde{\mathbf{x}}_t) - \frac{\mu}{1-\mu}(\tilde{\mathbf{x}}_t - \tilde{\mathbf{x}}_{t-1}) \\
&= -\frac{\eta}{K}\sum_{k=1}^{K}\mathbf{m}_{t+1}^{(k)} - \frac{\mu}{1-\mu}\frac{\eta}{K}\sum_{k=1}^{K}\mathbf{m}_{t+1}^{(k)} + \frac{\mu}{1-\mu}\frac{\eta}{K}\sum_{k=1}^{K}\mathbf{m}_t^{(k)} \\
&= -\frac{\eta}{(1-\mu)K}\sum_{k=1}^{K}(\mathbf{m}_{t+1}^{(k)} - \mu\mathbf{m}_t^{(k)}) \\
&= -\frac{\eta}{(1-\mu)K}\sum_{k=1}^{K}\nabla f_k(\mathbf{x}_t^{(k)}; \xi_t^{(k)}),
\end{aligned}
\tag{61}
$$

which exists for OLCO$_3$-OC, OLCO$_3$-VQ and OLCO$_3$-TC.

**Lemma 9.** *For OLCO$_3$ with Momentum SGD, we have*

$$\mathbb{E}\|\boldsymbol{m}_t^{(k)}\|_2^2 \le \frac{3(\sigma^2 + \kappa^2 + G^2)}{(1-\mu)^2}. \tag{62}$$

*Proof.*

$$
\begin{aligned}
\mathbb{E}\|\mathbf{m}_t^{(k)}\|_2^2 &= \mathbb{E}\|\sum_{t'=0}^{t-1}\mu^{t-1-t'}\nabla F_k(\mathbf{x}_{t'}^{(k)}; \xi_{t'}^{(k)})\|_2^2 = (\sum_{t'=0}^{t-1}\mu^{t-1-t'})^2\mathbb{E}\|\sum_{t'=0}^{t-1}\frac{\mu^{t-1-t'}}{\sum_{t'=0}^{t-1}\mu^{t-1-t'}}\nabla F_k(\mathbf{x}_{t'}^{(k)}; \xi_{t'}^{(k)})\|_2^2 \\
&\le (\sum_{t'=0}^{t-1}\mu^{t-1-t'})^2\mathbb{E}\|\nabla F_k(\mathbf{x}_{t'}^{(k)}; \xi_{t'}^{(k)})\|_2^2 \le \frac{3(\sigma^2 + \kappa^2 + G^2)}{(1-\mu)^2}.
\end{aligned}
\tag{63}
$$

$\square$

**Lemma 10.** *For OLCO$_3$ with Momentum SGD, we have*

$$\mathbb{E}\|\boldsymbol{p}_t\|^2 \le \frac{3\mu^2\eta^2(\sigma^2 + \kappa^2 + G^2)}{(1-\mu)^4}. \tag{64}$$

*Proof.*

$$
\begin{aligned}
\mathbb{E}\|\mathbf{p}_t\|^2 &= \frac{\mu^2}{(1-\mu)^2}\mathbb{E}\|\tilde{\mathbf{x}}_t - \tilde{\mathbf{x}}_{t-1}\|^2 = \frac{\mu^2\eta^2}{(1-\mu)^2}\mathbb{E}\|\frac{1}{K}\sum_{k=1}^{K}\mathbf{m}_t^{(k)}\|^2 \le \frac{\mu^2\eta^2}{(1-\mu)^2 K}\sum_{k=1}^{K}\mathbb{E}\|\mathbf{m}_t^{(k)}\|_2^2 \\
&\le \frac{3\mu^2\eta^2(\sigma^2 + \kappa^2 + G^2)}{(1-\mu)^4}.
\end{aligned}
\tag{65}
$$

$\square$

**Lemma 11.** *For OLCO$_3$-VQ with Momentum SGD and under Assumptions 2, 3, 4, and 5, the local error satisfies*

$$\mathbb{E}\|\boldsymbol{e}_t^{(k)}\|_2^2 \le \frac{12(1-\delta)}{(1-\mu)^2\delta^2}p^2\eta^2(\sigma^2 + \kappa^2 + G^2). \tag{66}$$

*Proof.* Let $S_t = \lfloor\frac{t}{p}\rfloor$,

$$
\begin{aligned}
\mathbb{E}\|\mathbf{e}_t^{(k)}\|_2^2 &= \mathbb{E}\|\mathbf{e}_{S_t p}^{(k)}\|_2^2 = \mathbb{E}\|\mathcal{C}(\Delta_{S_t p}^{(k)}) - \Delta_{S_t p}^{(k)}\|_2^2 \le (1-\delta)\mathbb{E}\|\Delta_{S_t p}^{(k)}\|_2^2 \\
&= (1-\delta)\mathbb{E}\|\sum_{t'=(S_t-1)p}^{S_T p-1}\eta\mathbf{m}_{t'}^{(k)} + \mathbf{e}_{(S_t-s-1)p}^{(k)}\|_2^2 \\
&\le (1-\delta)(1+\rho)\mathbb{E}\|\mathbf{e}_{(S_t-s-1)p}^{(k)}\|_2^2 + (1-\delta)(1+\frac{1}{\rho})\frac{3\eta^2 p^2(\sigma^2 + \kappa^2 + G^2)}{(1-\mu)^2}.
\end{aligned}
\tag{67}
$$

Therefore,

$$
\mathbb{E}\|\mathbf{e}_t^{(k)}\|_2^2 \le 3(1-\delta)(1+\frac{1}{\rho})\frac{p^2\eta^2(\sigma^2+\kappa^2+G^2)}{(1-\mu)^2} \sum_{i=0}^{\lfloor\frac{S_t}{s}\rfloor-1} [(1-\delta)(1+\rho)]^i
$$
$$
\le \frac{3(1-\delta)(1+\frac{1}{\rho})}{1-(1-\delta)(1+\rho)} \frac{p^2\eta^2(\sigma^2+\kappa^2+G^2)}{(1-\mu)^2}.
$$
(68)

Let $\rho = \frac{\delta}{2(1-\delta)}$ such that $1+\frac{1}{\rho} = \frac{2-\delta}{\delta} \le \frac{2}{\delta}$, then $\mathbb{E}\|\mathbf{e}_t^{(k)}\|_2^2 \le \frac{12(1-\delta)}{(1-\mu)^2\delta^2}p^2\eta^2(\sigma^2+\kappa^2+G^2)$. $\quad\square$

**Lemma 12.** *For OLCO$_3$-VQ with Momentum SGD and under Assumptions 1, 2, 3, 4, and 5, if the learning rate $\eta \le \frac{1-\mu}{\sqrt{72}L(s+1)p}$, we have*

$$
\frac{1}{KT}\sum_{t=0}^{T-1}\sum_{k=1}^{K}\mathbb{E}\|\mathbf{z}_t-\mathbf{x}_t^{(k)}\|_2^2 \le \frac{4\eta^2\sigma^2}{(1-\mu)^2}\Big(\frac{3}{(1-\mu)^2}+(s+1)p+\frac{12(1-\delta)(s+1)^2p^2}{\delta^2}\Big)
$$
$$
+\frac{12\eta^2\kappa^2}{(1-\mu)^2}\Big(\frac{1}{(1-\mu)^2}+(s+1)^2p^2+\frac{4(1-\delta)(s+1)^2p^2}{\delta^2}\Big)+\frac{12\eta^2G^2}{(1-\mu)^2}\Big(\frac{1}{(1-\mu)^2}+\frac{4(1-\delta)(s+1)^2p^2}{\delta^2}\Big).
$$
(69)

*Proof.* Let $S_t = \lfloor\frac{t}{p}\rfloor$,

$$
\frac{1}{K}\sum_{k=1}^{K}\mathbb{E}\|\mathbf{z}_t-\mathbf{x}_t^{(k)}\|_2^2 = \frac{1}{K}\sum_{k=1}^{K}\mathbb{E}\|\frac{1}{K}\sum_{k'=1}^{K}\Big(-\sum_{i=0}^{s-1}\Delta_{(S_t-i)p}^{(k')}-(\mathbf{x}_{S_tp}^{(k')}-\mathbf{x}_t^{(k')})\Big)
$$
$$
-\Big(-\sum_{i=0}^{s-1}\Delta_{(S_t-i)p}^{(k)}-(\mathbf{x}_{S_tp}^{(k)}-\mathbf{x}_t^{(k)})-\frac{1}{K}\sum_{k'=1}^{K}\mathbf{e}_{t-sp}^{(k')}\|_2^2
$$
$$
= \frac{1}{K}\sum_{k=1}^{K}\mathbb{E}\|\frac{1}{K}\sum_{k'=1}^{K}\Big(\eta\mathbf{m}_{(S_t-s)p}^{(k')}\sum_{\tau=0}^{t-1-(S_t-s)p}\mu^{\tau+1}+\sum_{t'=(S_t-s)p}^{t-1}\eta\nabla F_{k'}(\mathbf{x}_{t'}^{(k')};\xi_{t'}^{(k')})\sum_{\tau=0}^{t-1-t'}\mu^{\tau}\Big)
$$
$$
-\Big(\eta\mathbf{m}_{(S_t-s)p}^{(k)}\sum_{\tau=0}^{t-1-(S_t-s)p}\mu^{\tau+1}+\sum_{t'=(S_t-s)p}^{t-1}\eta\nabla F_k(\mathbf{x}_{t'}^{(k)};\xi_{t'}^{(k)})\sum_{\tau=0}^{t-1-t'}\mu^{\tau}\Big)
$$
$$
+\frac{1}{K}\sum_{k'=1}^{K}\mathbf{e}_{t-sp}^{(k')}+\frac{1}{K}\sum_{k'=1}^{K}\sum_{i=0}^{s-1}\mathbf{e}_{(S_t-i-s-1)p}^{(k')}-\sum_{i=0}^{s-1}\mathbf{e}_{(S_t-i-s-1)p}^{(k)}\|_2^2
$$
$$
\le \frac{3\eta^2}{K}\sum_{k=1}^{K}\mathbb{E}\|\frac{1}{K}\sum_{k'=1}^{K}\mathbf{m}_{(S_t-s)p}^{(k')}\sum_{\tau=0}^{t-1-(S_t-s)p}\mu^{\tau+1}-\mathbf{m}_{(S_t-s)p}^{(k)}\sum_{\tau=0}^{t-1-(S_t-s)p}\mu^{\tau+1}\|_2^2
$$
$$
+\frac{3\eta^2}{K}\sum_{k=1}^{K}\mathbb{E}\|\frac{1}{K}\sum_{k'=1}^{K}\sum_{t'=(S_t-s)p}^{t-1}\nabla F_{k'}(\mathbf{x}_{t'}^{(k')};\xi_{t'}^{(k')})\sum_{\tau=0}^{t-1-t'}\mu^{\tau}-\sum_{t'=(S_t-s)p}^{t-1}\nabla F_k(\mathbf{x}_{t'}^{(k)};\xi_{t'}^{(k)})\sum_{\tau=0}^{t-1-t'}\mu^{\tau}\|_2^2
$$
$$
+\frac{3}{K}\sum_{k=1}^{K}\mathbb{E}\|\frac{1}{K}\sum_{k'=1}^{K}\mathbf{e}_{t-sp}^{(k')}+\frac{1}{K}\sum_{k'=1}^{K}\sum_{i=0}^{s-1}\mathbf{e}_{(S_t-i-s-1)p}^{(k')}+\sum_{i=0}^{s-1}\mathbf{e}_{(S_t-i-s-1)p}^{(k)}\|_2^2.
$$
(70)

The first term

$$
\frac{3\eta^2}{K}\sum_{k=1}^{K}\mathbb{E}\|\frac{1}{K}\sum_{k'=1}^{K}\mathbf{m}_{(S_t-s)p}^{(k')}\sum_{\tau=0}^{t-1-(S_t-s)p}\mu^{\tau+1}-\mathbf{m}_{(S_t-s)p}^{(k)}\sum_{\tau=0}^{t-1-(S_t-s)p}\mu^{\tau+1}\|_2^2
$$
$$
\le \frac{3\eta^2}{K}\sum_{k=1}^{K}\mathbb{E}\|\mathbf{m}_{(S_t-s)p}^{(k)}\sum_{\tau=0}^{t-1-(S_t-s)p}\mu^{\tau+1}\|_2^2 \le \frac{3\eta^2}{(1-\mu)^2K}\sum_{k=1}^{K}\mathbb{E}\|\mathbf{m}_{(S_t-s)p}^{(k)}\|_2^2 \le \frac{9\eta^2(\sigma^2+\kappa^2+G^2)}{(1-\mu)^4},
$$
(71)

where the last inequality follows Lemma 9. Following Eq. (29), the second term can be bounded by

$$
\frac{3\eta^2}{K} \sum_{k=1}^{K} \mathbb{E} \| \frac{1}{K} \sum_{k'=1}^{K} \sum_{t'=(S_t-s)p}^{t-1} \nabla F_{k'}(\mathbf{x}_{t'}^{(k')}; \xi_{t'}^{(k')}) \sum_{\tau=0}^{t-1-t'} \mu^\tau - \sum_{t'=(S_t-s)p}^{t-1} \nabla F_k(\mathbf{x}_{t'}^{(k)}; \xi_{t'}^{(k)}) \sum_{\tau=0}^{t-1-t'} \mu^\tau \|_2^2
$$

$$
= \frac{3\eta^2}{K} \sum_{k=1}^{K} \mathbb{E} \| \sum_{t'=(S_t-s)p}^{t-1} [\frac{1}{K} \sum_{k'=1}^{K} (\nabla F_{k'}(\mathbf{x}_{t'}^{(k')}; \xi_{t'}^{(k')}) - \nabla f_{k'}(\mathbf{x}_{t'}^{(k')})) - (\nabla F_k(\mathbf{x}_{t'}^{(k)}; \xi_{t'}^{(k)}) - \nabla f_k(\mathbf{x}_{t'}^{(k)}))] \sum_{\tau=0}^{t-1-t'} \mu^\tau \|_2^2
$$

$$
+ \frac{3\eta^2}{K} \sum_{k=1}^{K} \mathbb{E} \| \sum_{t'=(S_t-s)p}^{t-1} (\frac{1}{K} \sum_{k'=1}^{K} \nabla f_{k'}(\mathbf{x}_{t'}^{(k')}) - \nabla f_k(\mathbf{x}_{t'}^{(k)})) \sum_{\tau=0}^{t-1-t'} \mu^\tau \|_2^2
$$

$$
\leq \frac{3\eta^2}{(1-\mu)^2 K} \sum_{k=1}^{K} \sum_{t'=(S_t-s)p}^{t-1} \mathbb{E} \| \nabla F_k(\mathbf{x}_{t'}^{(k)}; \xi_{t'}^{(k)}) - \nabla f_k(\mathbf{x}_{t'}^{(k)}) \|_2^2
$$

$$
+ \frac{3\eta^2}{(1-\mu)^2 K} (t - (S_t - s)p) \sum_{k=1}^{K} \sum_{t'=(S_t-s)p}^{t-1} \mathbb{E} \| \frac{1}{K} \sum_{k'=1}^{K} \nabla f_{k'}(\mathbf{x}_{t'}^{(k')}) - \nabla f_k(\mathbf{x}_{t'}^{(k)}) \|_2^2
$$

$$
\leq \frac{3\eta^2(s+1)p\sigma^2}{(1-\mu)^2} + \frac{3\eta^2(s+1)p}{(1-\mu)^2} \sum_{t'=t-(s+1)p}^{t-1} (\frac{6L^2}{K} \sum_{k=1}^{K} \mathbb{E} \| \mathbf{z}_{t'} - \mathbf{x}_{t'}^{(k)} \|_2^2 + 3\kappa^2),
$$

$$
\tag{72}
$$

where the last inequality follows Eq. (30). Combine the bounds of the first and second term with Lemma 11 and Eq. (31),

$$
\frac{1}{K} \sum_{k=1}^{K} \mathbb{E} \| \mathbf{z}_t - \mathbf{x}_t^{(k)} \|_2^2
$$

$$
\leq \frac{9\eta^2(\sigma^2+\kappa^2+G^2)}{(1-\mu)^4} + \frac{3\eta^2(s+1)p\sigma^2}{(1-\mu)^2} + \frac{3\eta^2(s+1)p}{(1-\mu)^2} \sum_{t'=t-(s+1)p}^{t-1} (\frac{6L^2}{K} \sum_{k=1}^{K} \mathbb{E} \| \mathbf{z}_{t'} - \mathbf{x}_{t'}^{(k)} \|_2^2 + 3\kappa^2)
$$

$$
+ \frac{3(s+1)}{K} \sum_{k=1}^{K} \sum_{i=0}^{s} \mathbb{E} \| \mathbf{e}_{(S_t-i-s)p}^{(k)} \|_2^2
$$

$$
\leq \frac{9\eta^2(\sigma^2+\kappa^2+G^2)}{(1-\mu)^4} + \frac{3\eta^2(s+1)p\sigma^2}{(1-\mu)^2} + \frac{9\eta^2(s+1)^2p^2\kappa^2}{(1-\mu)^2} + \frac{36(1-\delta)\eta^2(s+1)^2p^2(\sigma^2+\kappa^2+G^2)}{(1-\mu)^2\delta^2}
$$

$$
+ \frac{18\eta^2 L^2(s+1)p}{(1-\mu)^2} \sum_{t'=t-(s+1)p}^{t-1} \frac{1}{K} \sum_{k=1}^{K} \mathbb{E} \| \mathbf{z}_{t'} - \mathbf{x}_{t'}^{(k)} \|_2^2.
$$

$$
\tag{73}
$$

Sum the above inequality from $t = 0$ to $t = T - 1$ and divide it by $T$,

$$
\frac{1}{KT} \sum_{t=0}^{T-1} \sum_{k=1}^{K} \mathbb{E} \| \mathbf{z}_t - \mathbf{x}_t^{(k)} \|_2^2 \leq \frac{3\eta^2\sigma^2}{(1-\mu)^2} (\frac{3}{(1-\mu)^2} + (s+1)p + \frac{12(1-\delta)(s+1)^2p^2}{\delta^2})
$$

$$
+ \frac{9\eta^2\kappa^2}{(1-\mu)^2} (\frac{1}{(1-\mu)^2} + (s+1)^2p^2 + \frac{4(1-\delta)(s+1)^2p^2}{\delta^2}) + \frac{9\eta^2 G^2}{(1-\mu)^2} (\frac{1}{(1-\mu)^2} + \frac{4(1-\delta)(s+1)^2p^2}{\delta^2})
$$

$$
+ \frac{18\eta^2 L^2(s+1)^2p^2}{(1-\mu)^2} \frac{1}{KT} \sum_{t=0}^{T-1} \sum_{k=1}^{K} \mathbb{E} \| \mathbf{z}_t - \mathbf{x}_t^{(k)} \|_2^2.
$$

$$
\tag{74}
$$

If we choose $\eta \leq \frac{1-\mu}{\sqrt{72}L(s+1)p}$,

$$
\frac{1}{KT} \sum_{t=0}^{T-1} \sum_{k=1}^{K} \mathbb{E}\|\mathbf{z}_t - \mathbf{x}_t^{(k)}\|_2^2 \leq \frac{4\eta^2\sigma^2}{(1-\mu)^2}\left(\frac{3}{(1-\mu)^2} + (s+1)p + \frac{12(1-\delta)(s+1)^2p^2}{\delta^2}\right)
$$
$$
+ \frac{12\eta^2\kappa^2}{(1-\mu)^2}\left(\frac{1}{(1-\mu)^2} + (s+1)^2p^2 + \frac{4(1-\delta)(s+1)^2p^2}{\delta^2}\right) + \frac{12\eta^2 G^2}{(1-\mu)^2}\left(\frac{1}{(1-\mu)^2} + \frac{4(1-\delta)(s+1)^2p^2}{\delta^2}\right).
$$
(75)

$\square$

**Theorem 3.** *For OLCO$_3$-VQ with Momentum SGD and under Assumptions 1, 2, 3, 4, and 5, if the learning rate $\eta \leq \min\{\frac{1-\mu}{\sqrt{72}L(s+1)p}, \frac{1-\mu}{9L}\}$ and let $g(\mu, \delta, s, p) = \frac{15}{(1-\mu)^2} + \frac{60(1-\delta)(s+1)^2p^2}{\delta^2}$, then*

$$
\frac{1}{T} \sum_{t=0}^{T-1} \mathbb{E}\|\nabla f(\frac{1}{K} \sum_{k=1}^{K} \boldsymbol{x}_t^{(k)})\|_2^2 \leq \frac{6(1-\mu)(f(\boldsymbol{x}_0) - f_*)}{\eta T} + \frac{9L\eta\sigma^2}{(1-\mu)K}
$$
$$
+ \frac{4\eta^2 L^2}{(1-\mu)^2}[(4(s+1)p + g(\mu, \delta, s, p))\sigma^2 + (12(s+1)^2p^2 + g(\mu, \delta, s, p))\kappa^2 + g(\mu, \delta, s, p)G^2].
$$
(76)

*Proof.* Following the proof of Theorem 1 and the update rule Eq. (61), we have a similar inequality as Eq. (41) by choosing $\eta \leq \frac{1-\mu}{9L}$:

$$
\mathbb{E}_t f(\mathbf{z}_{t+1}) - f(\mathbf{z}_t)
$$
$$
\leq \frac{\eta}{1-\mu}\left(-\frac{1}{2}\|\nabla f(\mathbf{z}_t)\|_2^2 + \frac{L^2}{2K}\sum_{k=1}^{K}\|\mathbf{z}_t - \mathbf{x}_t^{(k)}\|_2^2\right) + \frac{L\eta^2}{2(1-\mu)^2}\left(\frac{3\sigma^2}{K} + \frac{3L^2}{K}\sum_{k=1}^{K}\|\mathbf{z}_t - \mathbf{x}_t^{(k)}\|_2^2 + 3\|\nabla f(\mathbf{z}_t)\|_2^2\right)
$$
$$
= -\frac{\eta}{2(1-\mu)}\left(1 - \frac{3L\eta}{1-\mu}\right)\|\nabla f(\mathbf{z}_t)\|_2^2 + \frac{L^2\eta}{2(1-\mu)K}\left(1 + \frac{3L\eta}{1-\mu}\right)\sum_{k=1}^{K}\|\mathbf{z}_t - \mathbf{x}_t^{(k)}\|_2^2 + \frac{3L\eta^2\sigma^2}{2(1-\mu)^2K}
$$
$$
\leq -\frac{\eta}{3(1-\mu)}\|\nabla f(\mathbf{z}_t)\|_2^2 + \frac{2\eta L^2}{3(1-\mu)K}\sum_{k=1}^{K}\|\mathbf{z}_t - \mathbf{x}_t^{(k)}\|_2^2 + \frac{3L\eta^2\sigma^2}{2(1-\mu)^2K}.
$$
(77)

Then for the averaged parameters $\frac{1}{K}\sum_{k=1}^{K}\mathbf{x}_t^{(k)}$,

$$
\|\nabla f(\frac{1}{K}\sum_{k=1}^{K}\mathbf{x}_t^{(k)})\|_2^2 \leq 2\|\nabla f(\frac{1}{K}\sum_{k=1}^{K}\mathbf{x}_t^{(k)}) - \nabla f(\mathbf{z}_t)\|_2^2 + 2\|\nabla f(\mathbf{z}_t)\|_2^2
$$
$$
\leq 2L^2\|\frac{1}{K}\sum_{k=1}^{K}\mathbf{e}_{t-sp}^{(k)} - \mathbf{p}_t\|_2^2 + 2\|\nabla f(\mathbf{z}_t)\|_2^2
$$
(78)
$$
\leq 4L^2\|\frac{1}{K}\sum_{k=1}^{K}\mathbf{e}_{t-sp}^{(k)}\|_2^2 + 4L^2\|\mathbf{p}_t\|_2^2 + 2\|\nabla f(\mathbf{z}_t)\|_2^2.
$$

Therefore

$$
\frac{1}{T}\sum_{t=0}^{T-1}\mathbb{E}\|\nabla f(\frac{1}{K}\sum_{k=1}^{K}\mathbf{x}_t^{(k)})\|_2^2 \leq \frac{6(1-\mu)[f(\mathbf{z}_0) - f(\mathbf{z}_T)]}{\eta T} + \frac{9L\eta\sigma^2}{(1-\mu)K} + \frac{4L^2}{KT}\sum_{t=0}^{T-1}\sum_{k=1}^{K}\mathbb{E}\|\mathbf{z}_t - \mathbf{x}_t^{(k)}\|_2^2
$$
$$
+ \frac{4L^2}{T}\sum_{t=0}^{T-1}\mathbb{E}\|\frac{1}{K}\sum_{k=1}^{K}\mathbf{e}_{t-sp}^{(k)}\|_2^2 + \frac{4L^2}{T}\sum_{t=0}^{T-1}\mathbb{E}\|\mathbf{p}_t\|_2^2
$$
$$
\leq \frac{6(1-\mu)(f(\mathbf{x}_0) - f_*)}{\eta T} + \frac{9L\eta\sigma^2}{(1-\mu)K}
$$
$$
+ \frac{4\eta^2 L^2}{(1-\mu)^2}[(4(s+1)p + g(\mu, \delta, s, p))\sigma^2 + (12(s+1)^2p^2 + g(\mu, \delta, s, p))\kappa^2 + g(\mu, \delta, s, p)G^2].
$$
(79)

where the last inequality follows Lemmas 10, 11 and 12 and $g(\mu, \delta, s, p) = \frac{15}{(1-\mu)^2} + \frac{60(1-\delta)(s+1)^2 p^2}{\delta^2}$. $\qquad\square$

## G  PROOF OF THEOREM 4

**Lemma 13.** *For OLCO$_3$-TC with Momentum SGD and under Assumptions 2, 3, 4, and 5, the local error satisfies*

$$\mathbb{E}\|\boldsymbol{e}_t^{(k)}\|_2^2 \le \frac{12(1-\delta)}{(1-\mu)^2\delta^2} p^2\eta^2(\sigma^2 + \kappa^2 + G^2). \tag{80}$$

*Proof.* Same as the proof of Lemma 11, except that $\mathbf{e}_{(S_t - s - 1)p}^{(k)}$ is replaced with $\mathbf{e}_{(S_t - 1)p}^{(k)}$. $\qquad\square$

**Lemma 14.** *For OLCO$_3$-TC with vanilla SGD and under Assumptions 2, 3, 4, and 5, the server error satisfies*

$$\mathbb{E}\|\boldsymbol{e}_t\|_2^2 \le \frac{96(2-\delta)(1-\delta)}{(1-\mu)^2\delta^4} p^2\eta^2(\sigma^2 + \kappa^2 + G^2). \tag{81}$$

*Proof.* Let $S_t = \lfloor\frac{t}{p}\rfloor$. Following the proof of Lemma 11 we have $\mathbb{E}\|\Delta_{S_t p}^{(k)}\|_2^2 \le \frac{3(1+\frac{1}{\rho})}{1-(1-\delta)(1+\rho)} \frac{p^2\eta^2(\sigma^2+\kappa^2+G^2)}{(1-\mu)^2}$. Therefore,

$$\mathbb{E}\|\boldsymbol{e}_t\|_2^2 \le 2(2-\delta)(1-\delta)(1+\frac{1}{\rho})\frac{1}{K}\sum_{k=1}^{K}\mathbb{E}\|\Delta_{S_t p}^{(k)}\|_2^2 + (1-\delta)(1+\rho)\mathbb{E}\|\boldsymbol{e}_{(S_t-1)p}\|_2^2$$

$$\le 2(2-\delta)(1-\delta)(1+\frac{1}{\rho})\frac{3(1+\frac{1}{\rho})}{1-(1-\delta)(1+\rho)}\frac{p^2\eta^2(\sigma^2+\kappa^2+G^2)}{(1-\mu)^2} + (1-\delta)(1+\rho)\mathbb{E}\|\boldsymbol{e}_{(S_t-1)p}\|_2^2$$

$$\le \frac{6(2-\delta)(1-\delta)(1+\frac{1}{\rho})^2}{[1-(1-\delta)(1+\rho)]^2(1-\mu)^2}p^2\eta^2(\sigma^2+\kappa^2+G^2), \tag{82}$$

where the first inequality follows the proof of Lemma 7. Let $\rho = \frac{\delta}{2(1-\delta)}$ such that $1 + \frac{1}{\rho} = \frac{2-\delta}{\delta} \le \frac{2}{\delta}$, then $\mathbb{E}\|\mathbf{e}_t^{(k)}\|_2^2 \le \mathbb{E}\|\mathbf{e}_t^{(k)}\|_2^2 \le \frac{96(2-\delta)(1-\delta)}{(1-\mu)^2\delta^4}p^2\eta^2(\sigma^2 + \kappa^2 + G^2)$. $\qquad\square$

**Lemma 15.** *For OLCO$_3$-TC with Momentum SGD and under Assumptions 1, 2, 3, 4, and 5, if the learning rate $\eta \le \frac{1-\mu}{\sqrt{72}L(s+1)p}$, we have*

$$\frac{1}{KT}\sum_{t=0}^{T-1}\sum_{k=1}^{K}\mathbb{E}\|\mathbf{z}_t - \mathbf{x}_t^{(k)}\|_2^2 \le \frac{3\eta^2\sigma^2}{(1-\mu)^2}\left(\frac{3}{(1-\mu)^2} + (s+1)p + 72h(\delta)p^2\right)$$

$$+ \frac{3\eta^2\kappa^2}{(1-\mu)^2}\left(\frac{3}{(1-\mu)^2} + 3(s+1)^2 p^2 + 72h(\delta)p^2\right) + \frac{3\eta^2 G^2}{(1-\mu)^2}\left(\frac{3}{(1-\mu)^2} + 72h(\delta)p^2\right), \tag{83}$$

*where $h(\delta) = \frac{1-\delta}{\delta^2}(1 + \frac{4(2-\delta)}{\delta^2})$.*

*Proof.* Let $S_t = \lfloor\frac{t}{p}\rfloor$,

$$\frac{1}{K}\sum_{k=1}^{K}\mathbb{E}\|\mathbf{z}_t - \mathbf{x}_t^{(k)}\|_2^2 = \frac{1}{K}\sum_{k=1}^{K}\mathbb{E}\|\frac{1}{K}\sum_{k'=1}^{K}\left(-\sum_{i=0}^{s-1}\mathcal{C}(\Delta_{(S_t-i)p}^{(k')}) - (\mathbf{x}_{S_t p}^{(k')} - \mathbf{x}_t^{(k')})\right)$$

$$- \left(-\sum_{i=0}^{s-1}\mathcal{C}(\Delta_{(S_t-i)p}^{(k)}) - (\mathbf{x}_{S_t p}^{(k)} - \mathbf{x}_t^{(k)}) - \frac{1}{K}\sum_{k'=1}^{K}\mathbf{e}_t^{(k)} - \mathbf{e}_{t-sp}\|_2^2, \tag{84}$$

where

$$\sum_{i=0}^{s-1} \mathcal{C}(\Delta_{(S_t-i)p}^{(k)}) + (\mathbf{x}_{S_tp}^{(k)} - \mathbf{x}_t^{(k)}) = \sum_{i=0}^{s-1} [\Delta_{(S_t-i)p}^{(k)} - \mathbf{e}_{(S_t-i)p}^{(k)}] + (\mathbf{x}_{S_tp}^{(k)} - \mathbf{x}_t^{(k)})$$

$$= \sum_{i=0}^{s-1} [\mathbf{m}_{(S_t-i-1)p}^{(k)} \sum_{\tau=0}^{p-1} \mu^{\tau+1} + \sum_{t'=(S_t-i-1)p}^{(S_t-i)p-1} \eta\nabla F(\mathbf{x}_{t'}^{(k)}; \xi_{t'}^{(k)}) \sum_{\tau=0}^{(S_t-i)p-1-t'} \mu^{\tau} + \mathbf{e}_{(S_t-i-1)p}^{(k)} - \mathbf{e}_{(S_t-i)p}^{(k)}]$$

$$+ \mathbf{m}_{S_tp}^{(k)} \sum_{\tau=0}^{t-1-S_tp} \mu^{\tau+1} + \sum_{t'=S_tp}^{t-1} \eta\nabla F_k(\mathbf{x}_{t'}^{(k)}; \xi_{t'}^{(k)}) \sum_{\tau=0}^{t-1-t'} \mu^{\tau}$$

$$= \mathbf{m}_{(S_t-s)p}^{(k)} \sum_{\tau=0}^{t-1-(S_t-s)p} \mu^{\tau+1} + \sum_{t'=(S_t-s)p}^{t-1} \eta\nabla F_k(\mathbf{x}_{t'}^{(k)}; \xi_{t'}^{(k)}) \sum_{\tau=0}^{t-1-t'} \mu^{\tau} + \mathbf{e}_{(S_t-s)p}^{(k)} - \mathbf{e}_{S_tp}^{(k)}.$$

$$(85)$$

Therefore,

$$\frac{1}{K} \sum_{k=1}^{K} \mathbb{E}\|\mathbf{z}_t - \mathbf{x}_t^{(k)}\|_2^2 = \frac{1}{K} \sum_{k=1}^{K} \mathbb{E}\|\frac{1}{K} \sum_{k'=1}^{K} \eta\mathbf{m}_{(S_t-s)p}^{(k')} \sum_{\tau=0}^{t-1-(S_t-s)p} \mu^{\tau+1} - \eta\mathbf{m}_{(S_t-s)p}^{(k)} \sum_{\tau=0}^{t-1-(S_t-s)p} \mu^{\tau+1}$$

$$+ \frac{1}{K} \sum_{k'=1}^{K} \sum_{t'=(S_t-s)p}^{t-1} \eta\nabla F_{k'}(\mathbf{x}_{t'}^{(k')}; \xi_{t'}^{(k')}) \sum_{\tau=0}^{t-1-t'} \mu^{\tau} - \sum_{t'=(S_t-s)p}^{t-1} \eta\nabla F_k(\mathbf{x}_{t'}^{(k)}; \xi_{t'}^{(k)}) \sum_{\tau=0}^{t-1-t'} \mu^{\tau}$$

$$+ \frac{1}{K} \sum_{k'=1}^{K} \mathbf{e}_{(S_t-s)p}^{(k')} - \mathbf{e}_{(S_t-s)p}^{(k)} + \mathbf{e}_{S_tp}^{(k)} + \mathbf{e}_{t-sp}\|_2^2$$

$$= \frac{3\eta^2}{(1-\mu)^2 K} \sum_{k=1}^{K} \mathbb{E}\|\mathbf{m}_{(S_t-s)p}^{(k)}\|_2^2 + \frac{3}{K} \sum_{k=1}^{K} \mathbb{E}\|\frac{1}{K} \sum_{k'=1}^{K} \mathbf{e}_{(S_t-s)p}^{(k')} - \mathbf{e}_{(S_t-s)p}^{(k)} + \mathbf{e}_{S_tp}^{(k)} + \mathbf{e}_{t-sp}\|_2^2$$

$$+ \frac{3\eta^2}{K} \sum_{k=1}^{K} \mathbb{E}\|\frac{1}{K} \sum_{k'=1}^{K} \sum_{t'=(S_t-s)p}^{t-1} \nabla F_{k'}(\mathbf{x}_{t'}^{(k')}; \xi_{t'}^{(k')}) \sum_{\tau=0}^{t-1-t'} \mu^{\tau} - \sum_{t'=(S_t-s)p}^{t-1} \nabla F_k(\mathbf{x}_{t'}^{(k)}; \xi_{t'}^{(k)}) \sum_{\tau=0}^{t-1-t'} \mu^{\tau}\|_2^2,$$

$$(86)$$

where the first term is bounded following Lemma 9 and the third term is bounded following Eq. (72). The second term

$$\frac{3}{K} \sum_{k=1}^{K} \mathbb{E}\|\frac{1}{K} \sum_{k'=1}^{K} \mathbf{e}_{(S_t-s)p}^{(k')} - \mathbf{e}_{(S_t-s)p}^{(k)} + \mathbf{e}_{S_tp}^{(k)} + \mathbf{e}_{t-sp}\|_2^2$$

$$\leq \frac{9}{K} \sum_{k=1}^{K} \mathbb{E}\|\mathbf{e}_{(S_t-s)p}^{(k)}\|_2^2 + \frac{9}{K} \sum_{k=1}^{K} \mathbb{E}\|\mathbf{e}_{S_tp}^{(k)}\|_2^2 + \frac{9}{K} \sum_{k=1}^{K} \mathbb{E}\|\mathbf{e}_{(S_t-s)p}\|_2^2 \qquad (87)$$

$$\leq \frac{1-\delta}{(1-\mu)^2\delta^2}(1 + \frac{4(2-\delta)}{\delta^2}) \cdot 216p^2\eta^2(\sigma^2 + \kappa^2 + G^2).$$

Combine these bounds,

$$\frac{1}{K} \sum_{k=1}^{K} \mathbb{E}\|\mathbf{z}_t - \mathbf{x}_t^{(k)}\|_2^2 \leq \frac{9\eta^2(\sigma^2 + \kappa^2 + G^2)}{(1-\mu)^4} + \frac{1-\delta}{(1-\mu)^2\delta^2}(1 + \frac{4(2-\delta)}{\delta^2}) \cdot 216p^2\eta^2(\sigma^2 + \kappa^2 + G^2)$$

$$+ \frac{3\eta^2(s+1)p\sigma^2}{(1-\mu)^2} + \frac{3\eta^2(s+1)p}{(1-\mu)^2} \sum_{t'=t-(s+1)p}^{t-1} (\frac{6L^2}{K} \sum_{k=1}^{K} \mathbb{E}\|\mathbf{z}_{t'} - \mathbf{x}_{t'}^{(k)}\|_2^2 + 3\kappa^2)$$

$$= \frac{18\eta^2 L^2(s+1)p}{(1-\mu)^2} \sum_{t'=t-(s+1)p}^{t-1} \frac{1}{K} \sum_{k=1}^{K} \mathbb{E}\|\mathbf{z}_{t'} - \mathbf{x}_{t'}^{(k)}\|_2^2$$

$$(88)$$

Sum the above inequality from $t = 0$ to $t = T - 1$, divide it by $T$, and choose $\eta \leq \frac{1-\mu}{\sqrt{72}L(s+1)p}$,

$$
\begin{aligned}
\frac{1}{KT} \sum_{t=0}^{T-1} \sum_{k=1}^{K} \mathbb{E}\|\mathbf{z}_t - \mathbf{x}_t^{(k)}\|_2^2 \leq{} & \frac{3\eta^2\sigma^2}{(1-\mu)^2}\left(\frac{3}{(1-\mu)^2} + (s+1)p + 72h(\delta)p^2\right) \\
& + \frac{3\eta^2\kappa^2}{(1-\mu)^2}\left(\frac{3}{(1-\mu)^2} + 3(s+1)^2p^2 + 72h(\delta)p^2\right) + \frac{3\eta^2G^2}{(1-\mu)^2}\left(\frac{3}{(1-\mu)^2} + 72h(\delta)p^2\right).
\end{aligned}
\tag{89}
$$

$\square$

**Theorem 4.** *For OLCO$_3$-TC with Momentum SGD and under Assumptions 1, 2, 3, 4, and 5, if the learning rate $\eta \leq \min\{\frac{1-\mu}{\sqrt{72}L(s+1)p}, \frac{1-\mu}{9L}\}$ and let $h(\delta) = \frac{1-\delta}{\delta^2}(1 + \frac{4(2-\delta)}{\delta^2})$, then*

$$
\begin{aligned}
\frac{1}{T} \sum_{t=0}^{T-1} & \mathbb{E}\|\nabla f(\frac{1}{K}\sum_{k=1}^{K}\boldsymbol{x}_t^{(k)})\|_2^2 \leq \\
\leq{} & \frac{6(1-\mu)(f(\boldsymbol{x}_0) - f_*)}{\eta T} + \frac{9L\eta\sigma^2}{(1-\mu)K} + \frac{6\eta^2L^2}{(1-\mu)^2}[\sigma^2(\frac{9}{(1-\mu)^2} + 2(s+1)p + 168h(\delta)p^2) \\
& + \kappa^2(\frac{9}{(1-\mu)^2} + 6(s+1)^2p^2 + 168h(\delta)p^2) + G^2(\frac{9}{(1-\mu)^2} + 168h(\delta)p^2)].
\end{aligned}
\tag{90}
$$

*Proof.* Following the proof of Theorem 3,

$$
\mathbb{E}_t f(\mathbf{z}_{t+1}) - f(\mathbf{z}_t) \leq -\frac{\eta}{3(1-\mu)}\|\nabla f(\mathbf{z}_t)\|_2^2 + \frac{2\eta L^2}{3(1-\mu)K}\sum_{k=1}^{K}\|\mathbf{z}_t - \mathbf{x}_t^{(k)}\|_2^2 + \frac{3L\eta^2\sigma^2}{2(1-\mu)^2K},
\tag{91}
$$

$$
\begin{aligned}
\|\nabla f(\frac{1}{K}\sum_{k=1}^{K}\mathbf{x}_t^{(k)})\|_2^2 \leq{} & 2\|\nabla f(\frac{1}{K}\sum_{k=1}^{K}\mathbf{x}_t^{(k)}) - \nabla f(\mathbf{z}_t)\|_2^2 + 2\|\nabla f(\mathbf{z}_t)\|_2^2 \\
\leq{} & 2L^2\|\frac{1}{K}\sum_{k=1}^{K}\mathbf{e}_t^{(k)} + \mathbf{e}_{t-sp} - \mathbf{p}_t\|_2^2 + 2\|\nabla f(\mathbf{z}_t)\|_2^2 \\
\leq{} & 6L^2\|\frac{1}{K}\sum_{k=1}^{K}\mathbf{e}_{t-sp}^{(k)}\|_2^2 + 6L^2\|\mathbf{e}_{t-sp}\|_2^2 + 6L^2\|\mathbf{p}_t\|_2^2 + 2\|\nabla f(\mathbf{z}_t)\|_2^2.
\end{aligned}
\tag{92}
$$

Therefore

$$
\begin{aligned}
\frac{1}{T} \sum_{t=0}^{T-1} & \mathbb{E}\|\nabla f(\frac{1}{K}\sum_{k=1}^{K}\mathbf{x}_t^{(k)})\|_2^2 \leq \frac{6(1-\mu)[f(\mathbf{z}_0) - f(\mathbf{z}_T)]}{\eta T} + \frac{9L\eta\sigma^2}{(1-\mu)K} + \frac{4L^2}{KT}\sum_{t=0}^{T-1}\sum_{k=1}^{K}\mathbb{E}\|\mathbf{z}_t - \mathbf{x}_t^{(k)}\|_2^2 \\
& + \frac{6L^2}{T}\sum_{t=0}^{T-1}\mathbb{E}\|\frac{1}{K}\sum_{k=1}^{K}\mathbf{e}_{t-sp}^{(k)}\|_2^2 + \frac{6L^2}{T}\sum_{t=0}^{T-1}\mathbb{E}\|\mathbf{e}_{t-sp}\|_2^2 + \frac{6L^2}{T}\sum_{t=0}^{T-1}\mathbb{E}\|\mathbf{p}_t\|_2^2 \\
\leq{} & \frac{6(1-\mu)(f(\mathbf{x}_0) - f_*)}{\eta T} + \frac{9L\eta\sigma^2}{(1-\mu)K} + \frac{6\eta^2L^2}{(1-\mu)^2}[\sigma^2(\frac{9}{(1-\mu)^2} + 2(s+1)p + 168h(\delta)p^2) \\
& + \kappa^2(\frac{9}{(1-\mu)^2} + 6(s+1)^2p^2 + 168h(\delta)p^2) + G^2(\frac{9}{(1-\mu)^2} + 168h(\delta)p^2)],
\end{aligned}
\tag{93}
$$

where the last inequality follows Lemmas 10, 13, 14 and 15. $\square$

