# OpenReview forum: "Delay-Tolerant Local SGD for Efficient Distributed Training"
_ICLR.cc/2021/Conference — Reject_

### Official Review · AnonReviewer4 · 2020-10-21
**Some major issues with the novelty and experimental setups**

**Rating:** 4
**Confidence:** 4

**Review:**

--- Summary

This paper proposes OLCO3, a new delay-tolerant SGD communication scheme and training framework for distributed deep neural network training. OLCO3 combines the existing ideas of Stale synchronous Parallel, batching the communication of doing multiple iterations, and gradient compression to achieve more communication efficiency. OLCO3 also uses staleness compensation and compression compensation techniques to improve model convergence under high stateness. Theoretical analysis shows that OLCO3 converges under SGD and momentum SGD. The evaluation was done with ResNet models on Cifar-10 and ImageNet datasets. Under a high staleness delay tolerance 56, OLCO3 achieves better convergence and has lower communication traffic than the baseline methods.


--- Strengths

- The paper is clearly written and has theoretical grounds.


--- Major issues

My major issue with this paper is its novelty and experimental setups.

- OLCO3 simply applies three existing delay-tolerant SGD techniques together. I agree that sometimes it needs special designs to combine multiple techniques and that successfully combining them could be an important contribution, but I am not sure it is the case for this work. For example, [Cui et al., 2014], used Stale synchronous Parallel and batched the communication for multiple iterations. [Lin et al., 2018] used gradient compression, and they also accumulated all the compression residuals and added them together with the gradients of the next iteration. So OLCO3's communication scheme and compression compensation techniques are not really novel.

* [Cui et al., 2014] Exploiting bounded staleness to speed up big data analytics. USENIX ATC 2014.
* [Lin et al., 2018] Deep Gradient Compression: Reducing the Communication Bandwidth for Distributed Training. ICLR 2018.

- The experiments in this paper used a delay tolerance of 56, which I think is too high for ResNet training. A delay tolerance of 56 means the workers will only communicate every 56 iterations when s=1, or they will have to tolerate 56 iterations of stale data when p=1. The ResNet-50 model has less than 25 million model parameters, and when represented in float32, they are only 100 MB in size. The sizes of the gradients are the same as the model parameters, so they are also 100 MB. I don't think it makes a lot of sense to delay the communication of just 100 MB of data for 56 iterations, especially when there are only two or four 4-GPU machines connected with 40 Gbps Ethernet. Showing that OLCO3 outperforms the other methods under this unrealistic delay tolerance is not very meaningful. The paper mentions that the high delay tolerance could be useful to federated learning, but federated learning is not very common right now, and the paper's experiments are not performed on a federated learning setup either.

- The evaluation also did not measure the wall-clock training time. So it is not clear to me 1) how using the more stale gradient updates speeds up the training and 2) how much computational overhead the gradient compression incurs. I suggest the authors compare the wall-clock training time of their system with the wall-clock training time of synchronous SGD using the state-of-the-art distributed training frameworks.


--- Other comments

- Section 2, second paragraph, "Note that Pipe-SGD is different from asynchronous SGD (Ho et al., 2013; Lian et al., 2015) which computes stochastic gradient using stale model and does not parallelize the computation and communication of a worker."
This statement is not correct. The State Synchronous Parallel parameter server proposed by [Ho et al., 2013] and [Cui et al., 2014] actually pipelines the communication with computation. Also, whether the communication is pipelined with the computation or not is orthogonal to the communication scheme. For example, the communication can be pipelined with the computation even for synchronous SGD because the model parameter updates from the training backward pass come out layer by layer, and the parameter server systems usually send out the computed parameter updates of the upper layers before the updates of the lower layers are computed.

- Section 5, experimental setup.
Why was ResNet-50 model used for the ImageNet dataset but a larger ResNet-110 model was used for the much smaller Cifar-10 dataset?

- Figure 1.
Please explain whether the communication budget is the total budget over all workers or the per-worker budget. The graph shows that Cifar-10 only has less than 1 GB of communication traffic even for the baseline synchronous SGD model. If that's the case, it is arguable whether it is indeed useful to do gradient compression when the total traffic for the whole training is only 1 GB.

---

### Official Review · AnonReviewer3 · 2020-10-27
**Delay-Tolerant Local SGD for Efficient Distributed Training**

**Rating:** 5
**Confidence:** 3

**Review:**

Review: This paper studies distributed training of neural networks. The major obstacles in distributed training are communication costs and communication delays. In the literature there exists different methods which attempt to overcome these two issues but, as far as the authors claim, none of the existing algorithms succeeds in dealing with both these aspects at the same time. The authors propose a novel distributed method OLCO_3 which is designed to address both communication costs and communication delays. In particular, the authors propose two variants of the method: OLCO_3 TC which comprises pipelining and compensation, and OLCO_3 VQ which comprises pipelining with communication-dependent compressor. Both the versions of OLCO_3 are analyzed from a theoretical perspective: under some assumptions the authors conduct a theoretical analysis of the convergence of the proposed schemes. Finally, the method in its two variants is benchmarked and compared with state-of-the-art distributed algorithms.


+ The authors makes a comprehensive review of the literature and the major techniques used in distributed training of NNs.

+ The authors are focusing on two really critical aspects of distributed training: communication costs and delays. These two aspects represent the bottleneck of distributed training and contributes in these directions would be of great impact.


Concerns:

- The paper is not well-written. In addition to some typos, the style is confused and therefore the paper results hard to read. The content is not clearly explained and presented. Overall the paper requires some re-writing and polishing.


- The theoretical results are not commented enough and the derived bounds do require some extra explanation and contextualization.


- Since the OLCO_3 method is motivated by the authors in terms of communication and delays efficiency, the theoretical analysis should also maybe account more for these two aspects and underline the major advantages with respect to the state-of-the-art methods in terms of communication costs and delays handling.


- Regarding the benchmark section, only test metrics are shown but test metrics are not directly related to the convergence but rather with the generalization properties, while the theoretical results focus on the convergence.

---------------------------------------------------------------------------------------------------------------------------------------------------

---

### Official Review · AnonReviewer2 · 2020-10-29
**The novelty is not too high. Comparison with more advanced baselines is needed.**

**Rating:** 5
**Confidence:** 4

**Review:**

This paper proposes a method, called OLCO3, to reduce the communication cost in distributed learning. Experiments on real datasets are used for evaluation. The paper is well written.

The main idea of OLCO3 is to combine many existing communication reduction methods, including pipelining, gradient compression and periodic averaging. There does exist some novelty in OLCO3, but the novelty is not high. Furthermore, there has appeared one similar paper[A] which combines sparsification, quantization and local SGD into the same framework for communication reduction. But this paper does not cite [A], and empirical comparison with [A] is also not provided.

Another shortcoming of OLCO3 is that the computation-communication overlapping technique will introduce an extra memory cost of O(sd), which might be unacceptable for large deep models with a huge d when s is relatively large.

For experiments, the convincingness can be improved if test accuracy/training error vs. wall-clock time is also provided.

[A]. Basu, Debraj, et al. "Qsparse-local-SGD: Distributed SGD with quantization, sparsification and local computations." Advances in Neural Information Processing Systems. 2019.


------------------------
After discussion:

The authors do not provide rebuttal. Hence, I keep the original opinion to give this paper a weak reject.

---

### Official Review · AnonReviewer5 · 2020-11-06
**initial review**

**Rating:** 5
**Confidence:** 4

**Review:**

The paper considers delay-tolerant and communication-efficient in distributed training and proposes a training framework OLCO3 with local update steps, staleness compensation, and compression compensation. The proposed OLCO3 can be generalized to OLCO3-TC & OLCO3-OC (master-slave case), and OLCO3-VQ (both master-salve and all-reduce case).

### pros.
* the paper is well-written; the arguments are supported by both empirical and theoretical results.
* the convergence analyses are provided for OLCO3-VQ and OLCO3-TC, for both SGD and momentum SGD.
* in numerical results, both iid and non-iid cases are considered.

### cons.
* missing compression compensation baseline. the paper considers local update, staleness compensation, and compression compensation, but in the empirical results, only local SGD and some delay tolerance methods are considered. it is suggested to also include the results from the powerSGD and the signSGD, thus the readers can identify the source of quality loss.
* different compression operators are used for different OLCO3 variants. I noticed that in the evaluation part, the paper considers  OLCO3-OC with signSGD, OLCO3-VQ with powerSGD, and OLCO3-TC with signSGD. it is encouraged to justify such a design choice.
* unclear practical impact. even though it is intuitive to design a training system that has local update steps, staleness compensation, and compression compensation, its practical impact is still unclear to me (due to the trade-off between test accuracy and system performance). it is encouraged to include (e.g. a simulated) results to illustrate the potential trade-off (e.g. time-to-accuracy) on different distributed training scenarios (e.g. differ in latency, bandwidth, local computation capability).
* the ideas like local update steps, staleness compensation, and compression compensation, have been well developed in the distributed machine learning community. though I do acknowledge the efforts of formulating/combining these ideas into a unified framework, the significance (novelty) of the paper might still have some limitations (as the proof seems quite standard to me). I would like to encourage authors to provide comprehensive empirical results to justify the pros and cons of the proposed scheme, as well as some practical guidelines.

---

### Decision · Program_Chairs · 2021-01-07
**Final Decision**

**Decision:**

Reject

**Comment:**

No discussion or answers to concerns are offered by the authors.
Given this, the current consensus remains the same as the initial review status, and AC's meta-review cannot provide any additional information.

This leads to rejection